# Hyperparameter tuning framework for calibrating analytical wake models using SCADA data of an offshore wind farm applied on FLORIS

Diederik van Binsbergen[1,2], Pieter-Jan Daems[1], Timothy Verstraeten[1], Amir R. Nejad[2], and Jan Helsen[1]

[1]Department of Mechanical Engineering, Vrije Universiteit Brussel, Pleinlaan 2, Brussels, 1050, Belgium
[2]Department of Marine Technology, NTNU, Jonsvannsveien 82, Trondheim, 7050, Norway

**Correspondence:** Diederik van Binsbergen (dirk.w.van.binsbergen@ntnu.no)

**Abstract.** This work presents a robust methodology for calibrating analytical wake models, as demonstrated on the velocity deficit parameters of the Gauss-Curl Hybrid model using four years of time series SCADA data from an offshore wind farm, with a Tree-Structured Parzen Estimator employed as a sampler. Initially, a sensitivity analysis of wake parameters and their linear correlation is conducted. The wake model is used with a turbulence intensity of 0.06, and no blockage model is considered. Results show that the tuning parameters that are multiplied with the turbine-specific turbulence intensity pose higher sensitivity than tuning parameters not giving weight to the turbulence intensity. It is also observed that the optimization converges with a higher residual error when inflow wind conditions are affected by neighbouring wind farms. The significance of this effect becomes apparent when the energy yield of turbines situated in close proximity to nearby wind farms is compared. Sensitive parameters show strong convergence, while parameters with low sensitivity show significant variance after optimization. Additionally, coastal influences are observed to affect the calibrated results, with wind from land leading to faster wake recovery than wind from the sea. Given the assumption of constant turbulence intensity in this work, recalibration is required when more representative site-specific turbulence intensity measurements are used as input to the model. Caution is advised when using these results without considering underlying model assumptions and site-specific characteristics, as these findings may not be generalizable to other locations without further recalibration.

## 1 Introduction

The wind energy sector is experiencing significant growth driven by the demand for renewable energy for the world's energy needs. According to the International Energy Agency (IEA (2022)), the amount of electricity generation from wind increased by 17% in 2021, which is an increase in growth of 55% compared to 2020. The pursuit of growth has led to a significant increase in the size of wind turbines and wind farms, capitalizing on economies of scale, particularly in operation and maintenance. This increase in wind turbine and wind farm size poses new challenges, among others, related to flow physics, as highlighted in studies by Veers et al. (2019); Porté-Agel et al. (2019); Meyers et al. (2022); Veers et al. (2023). Wind turbine wakes are typically characterized by a velocity deficit and increased turbulence behind the wind turbine (Lissaman (1979)). The velocity deficit can result in considerable power losses in downwind turbines, while the added turbulence leads to increased fatigue

loads (Thomsen and Sørensen (1999); van Binsbergen et al. (2020); Nejad et al. (2022); Verstraeten et al. (2019)). Although it is widely acknowledged that clustered wind turbines lead to reduced power production for downwind turbines, the exact degree of these losses remains uncertain, especially in the context of growing wind turbine and wind farm size.

Methods that optimize the power production of the wind farm, such as layout optimization (Baker et al. (2019); Sickler et al. (2023)), axial induction control (Annoni et al. (2015); Dilip and Porté-Agel (2017); Kheirabadi and Nagamune (2019); Bossanyi and Ruisi (2021)), wake steering (Fleming et al. (2019); Quick et al. (2020); Kheirabadi and Nagamune (2019); Doekemeijer et al. (2021)) and power set-point optimization (Verstraeten et al. (2021)), have the potential to reduce the levelized cost of energy. However, given the considerable complexity of wakes and the stochastic nature of wind, a significant degree of uncertainty remains within the field of layout optimization and farm control. As a result of this, wind power plant flow physics has been recognized as a significant challenge for the future, as outlined by Veers et al. (2019, 2023).

Wind turbine wakes are analyzed and modeled on different levels of fidelity for different purposes. Engineering tools such as the FLow Redirection and Induction in Steady-state (FLORIS) framework by NREL (2023) and the PyWake simulation tool by DTU (2023) are used to study the interaction between turbines within a wind farm and the consequences on power production in a low-fidelity but computationally inexpensive way. Both FLORIS and PyWake consist of various wake models, aiming to accurately simulate the interaction between multiple turbines within a wind farm and can be used for wind farm design, control, and optimization. This study employs the Gauss-Curl Hybrid (GCH) model, as described in King et al. (2021), within the FLORIS framework. However, the methodology is not restricted to specific models and frameworks. As understanding the historical evolution of wake models is essential to understanding the foundations of the chosen model, it will be discussed in the next section.

## 1.1 Wake model evolution

Analytical wake models within engineering frameworks are generally subdivided into four submodels: the wake velocity deficit model, the wake deflection model, the wake-added turbulence model, and the wake combination model. In recent years significant progress has been made, both in scientific comprehension of the physics that are in play, and the modeling of these physical phenomena. The propagation of the wake velocity deficit is the reason power losses occur for clustered wind farms. The wake propagation can be subdivided into a near-wake and a far-wake region. For the near-wake region, the wake mixing is mainly dominated by the wake-added turbulence of the wind turbine and the tip vortices are still present within the flow, while for the far-wake region wake mixing is mainly dominated by mixing due to atmospheric turbulence Sanderse (2009). As previously mentioned, the wake recovery due to mixing is heavily dependent on atmospheric conditions. Over the years, a range of wake models have been developed, such as the Jensen model (Jensen (1983); Katić et al. (1987)), Gaussian-shaped models (Bastankhah and Porté-Agel (2014, 2016); Niayifar and Porté-Agel (2015); Blondel and Cathelain (2020); Zong and Porté-Agel (2020)), and the Cumulative Curl (Blondel and Cathelain (2020); Bastankhah et al. (2021); Bay et al. (2022)) and TurbOPark (Nygaard et al. (2020); Pedersen et al. (2022); Nygaard et al. (2022)) models motivated by Ørsted (2019). Each model describes the velocity deficit in a unique way and many of these models have found integration within frameworks such as FLORIS and PyWake.

The Jensen model is a longstanding reliable analytical wake model based on the conservation of mass, correlating the wind speed behind the rotor with the thrust coefficient. The Jensen model is a top-hat model, meaning that the Jensen model assumes a constant velocity across a wake cross-section. Furthermore, the model assumes linear wake growth and wake decay proportional to the inverse of the downwind distance. The top-hat model results in unrealistically sensitive power predictions downwind and overestimates the velocity deficit at the edge of the wake while underestimating it in the center. This resulted in the development of a new model by Bastankhah and Porté-Agel (2014, 2016); Niayifar and Porté-Agel (2015), which follows a self-similar Gaussian distribution. This model, recognized as the Gauss model, consists of four tuning parameters: the ones related to wake expansion ($k_a$ and $k_b$), and the ones which define the transition point from the near-wake region to the far-wake region ($\alpha$ and $\beta$). In research conducted by King et al. (2021), analytical modifications were made to the Gaussian model by adding the effect of curled wakes, as depicted in the Curl model by Martínez-Tossas et al. (2019), and implementing secondary-steering effects, observed by Fleming et al. (2018); Wang et al. (2018). This model is known as the GCH model.

Over the years, it has become clear that traditional wake models often underestimate wake losses in the far-wake region, thereby overestimating the expected yield, as per Ørsted (2019). This triggered the development of the Cumulative Curl model by Blondel and Cathelain (2020); Bastankhah et al. (2021); Bay et al. (2022) and the TurbOPark model by Nygaard et al. (2020); Pedersen et al. (2022). While the Cumulative Curl model builds upon the advancements from the Gaussian wake model, the TurbOPark model developed by Nygaard et al. (2020) originates the Jensen/Park model. The advancements made by Pedersen et al. (2022) incorporate the Gaussian deficit profile.

## 1.2 Calibration of analytical wake models

Within the development of analytical wake models, calibration of parameters is necessary. Not calibrating scaling parameters can potentially result in over-or underestimation of the energy yield and suboptimal wind turbine siting within a wind farm if used for the design of a wind farm. This calibration can initially be carried out by comparing the analytical wake models with high-fidelity computational fluid dynamics (CFD) models, large eddy simulation (LES) models (Gebraad et al. (2014); Fleming et al. (2017); Doekemeijer et al. (2019); Zhang and Zhao (2020); Doekemeijer et al. (2020); Bay et al. (2022)) or wind tunnel experiments (Sanderse et al. (2022); Campagnolo et al. (2022)). This comparison provides a general idea of the parameter value. However, given the differences in site-specific factors (like the topography, wind resource, surface roughness, atmospheric stability, turbulence intensity, and general gradients), wind farm-specific attributes (such as size and spacing), and wind turbine-specific properties (like power-thrust curves), wind farm specific calibration is required to improve accuracy of the acquired results. The recent study by Göçmen et al. (2022) further highlights the importance of an appropriate calibration procedure for control-oriented models. Alternatives for calibration, such as field measurements (Fuertes et al. (2018); Cañadillas et al. (2022)), are available but require the installation of additional equipment.

Calibrating wake models on SCADA data has the advantage that the site-specific, farm-specific, and turbine-specific uncertainties can be minimized through optimization, and is currently being used for farm based control. For example, work done by Göçmen and Giebel (2018); Teng and Markfort (2020); Schreiber et al. (2020); van Beek et al. (2021); Göçmen et al. (2022) all use SCADA data to calibrate parameters that dictate the wake model performance.

In van Beek et al. (2021) a sensitivity study is performed on the parameters of the Gauss-Curl-Hybrid model, described in King et al. (2021), and concluded that the model is overparameterized. This can result in ill-posed and non-uniqueness problems, as demonstrated by Doekemeijer et al. (2022), where the so-called 'waterbed effect' occurs between the freestream turbulence and the wind direction variability within the FLORIS framework, making it impossible to identify the right value for a set of parameters. To counteract this, Schreiber et al. (2020) applied a singular value decomposition to remove correlation and overparameterization by mapping the original parameters onto a new set of uncorrelated parameters. However, in Göçmen et al. (2022) blind tests were carried out, where for a similar framework, different resultant parameter sets were obtained. Furthermore, Göçmen et al. (2022) stated that the turbine performance in yaw is highly uncertain. This can further result in non-uniqueness issues. In light of these findings, an approach is developed that provides a consistent optimization framework, emphasizing robust filtering, optimization, and validation. By integrating wind speed and wind direction into the optimization, potential biases in determining freeflow conditions are effectively mitigated. The use of time series data in calibration ensures that atmospheric inflow biases are not irreversibly categorized, preserving the potential for subsequent post-processing and ad-hoc analysis. Each optimization of individual timestamps within the SCADA data operates independently, ensuring a smooth parameter distribution across varied wind speeds and wind directions while limiting the effect of sporadic outliers. By incorporating a sensitivity study and energy ratio comparisons, thorough validation of determined hyperparameters is ensured. Assessing the Pearson correlation ensures minimized correlation among tuning parameters, thereby mitigating the occurrence of overparametrization.

## 1.3   Challenges in calibration of analytical wake models with SCADA data

While the correlation between wake parameters is one source of uncertainty within the framework of calibrating wake parameters of analytical wake models, additional sources of uncertainty influence the effectiveness of the calibration framework.

A first additional source of uncertainty arises from the estimation of the freeflow wind speed, wind direction, and nacelle direction. Estimating the freeflow wind speed and wind direction of a wind farm can be difficult due to stochastic and sensor uncertainty. Stochastic uncertainty covers all variations or random fluctuations in wind characteristics, ranging from turbulence, evolving weather patterns, or diurnal cycles. Specifically, turbulence can introduce uncertainty within averaged intervals, while shifts in weather and daily patterns might cause the mean value of the parameter to drift. Barthelmie et al. (2009) points out the complexities concerning the estimation of the freeflow wind speed and wind direction. Since wind speed serves as the most sensitive parameter for the active power of wind turbines below rated, achieving precise estimates is essential when comparing wake models with measurements. The freeflow wind speed can be calculated using the wind speed of the freeflow wind turbines or using a wind mast. Estimating wind speed from active power for calibration could yield more accurate results compared to using the nacelle anemometer, primarily due to the large sensitivity to rapid changes in wind speed of the nacelle anemometer. These rapid changes in wind speed do not agree with the assumption of steady-state inflow within the analytical wake model frameworks. Furthermore, wind direction measurements can display a bias of up to $5°$, even in the case of well-maintained wind vanes, as per Barthelmie et al. (2009). Moreover, individual wind turbines can have distinct biases in wind direction measurements. Determining the wind direction can be fundamentally more complex due to the non-linear relation between the

wind direction and the active power of wind turbines within a wind farm. An example of direction calibration towards true north based on energy ratios can be found in Doekemeijer et al. (2022).

Secondly, wind speed and turbulence gradients due to external wakes and terrain effects pose a challenge. Barthelmie et al. (2007) investigated the effect of coastal wind speed gradients and concluded that these gradients must be considered when optimizing wake models for wind farms located in coastal regions. Doekemeijer et al. (2022) mentioned that the effect of higher turbulence from the coast, compared to the sea, can induce more wake recovery. In order to address the heterogeneous inflow, Göçmen et al. (2022) created a non-homogeneous flow field for wind speed and turbulence intensity by using anemometer data for calibration, while Schreiber et al. (2020) applied spatial correction factors on the wind speed. With an increase in the number of wind farms being built, farm-to-farm effects are becoming significantly more frequent. Pettas et al. (2021) showed that the external wake effects result in increased turbulence intensity and structural loading, with a reduced wind speed for the considered wind farm.

In addition, time averaging of SCADA data is often necessary to remove short-term fluctuations and noise in the data, which is especially present for anemometer measurements. The disadvantage of averaging SCADA data is the loss of information. The averaged timestamp can be subject to changes in wind speed or wind direction, which can make the resulting timestamp not representative to the analytical wake models. Therefore it is important to not only consider the average of the timestamp, but also higher-order statistical moments, like variance.

Furthermore, spatial and temporal variability across the wind farm introduces complexity. Many wake models assume steady and horizontally homogeneous wind inflow. This assumption will always introduce additional uncertainty. Therefore it is of significant importance to carefully filter the data used for the purpose of calibration. Especially with the increasing size of wind farms, the assumption that each wind turbine within the wind farm experiences the same wind condition at one point in time is not valid anymore. Specialized frameworks, like FLORIDyn by Becker et al. (2022), have been developed specifically to address the temporal and spatial variability across a wind farm. For steady-state models, manipulation of the SCADA data should be considered, incorporating a time lag for specific wind conditions, in line with the methodology carried out in Ávila et al. (2023).

Additionally, uncertainty originating from natural fluctuations, like diurnal and annual cycles, should be considered, since these affect atmospheric stability. Measuring atmospheric stability itself is prone to uncertainty, often arising from sensor-related inaccuracies and stochastic variations. In work done by Hansen et al. (2011), they revealed that stable atmospheric conditions, characterized by low turbulence, correlated with larger power deficits, than unstable atmospheric conditions due to limited flow mixing. Wang et al. (2022) showed the effect of atmospheric stability due to diurnal cycles on the internal wake patterns. They concluded, similarly, that a stable atmosphere during the night resulted in larger wake losses than an unstable atmosphere during the day. This implies that there should be a clear distinction between stable and unstable atmospheric conditions when calibrating wind turbine wake models.

Lastly, recent advances in the understanding of flow physics in wind farms have highlighted the issue of wind farm blockage (Porté-Agel et al. (2019); Meyers et al. (2022)), where Bleeg et al. (2018) observed that the blockage effect results in less energy generation than initially expected for front-row wind turbines. The deceleration caused by the blockage will subsequently result

in a deflection upwards and sideways due to mass conservation (Porté-Agel et al. (2019)). Furthermore, studies by Wu and Porté-Agel (2017); Allaerts and Meyers (2017); Schneemann et al. (2021) showed that the global blockage effect is strongly influenced by atmospheric stability. Recent advancements in modeling the effects of wind farm blockage effects (Branlard and Forsting (2020); Branlard et al. (2020); Nygaard et al. (2020), have facilitated the reduction of uncertainty in wind farm construction planning, as implemented by Munters et al. (2022); Nygaard et al. (2022).

## 1.4 Objectives

The main objective of this study is to perform a model calibration using SCADA data on the wake velocity deficit parameters of the GCH wake model, while maintaining homogeneous freeflow conditions within the FLORIS framework. This optimization is performed on bottom-fixed wind turbines within a large offshore wind farm. The intention of this study is to set a constructive foundation for the calibration of wake parameters, creating the opportunity for possible advancements in the future. The optimization operates under the assumption that all wind turbines are perfectly aligned with the direction of the freeflow wind. No wind speed or turbulence gradients are introduced to the flow field. SCADA data is averaged into 10 minute averages and no alternations are made to account for temporal variability across the wind farm. A constant turbulence intensity is used within this optimization framework, as variable turbulence intensity introduces additional sensor and model uncertainties. It is furthermore observed by Doekemeijer et al. (2020) that the turbulence intensity within the FLORIS framework does not fully represent the physical turbulence intensity. While atmospheric stability significantly influences the results obtained, it is not analyzed in this study. Additionally, the model does not account for wind farm blockage effects, as it does not include neighbouring wind farms. This assumption is considered acceptable, given the limited indications of spatially varying wind directions attributable to blockage at this wind farm. Since it is assumed that all wind turbines are perfectly aligned, optimization of wake deflection parameters is not considered. This is in line with van Beek et al. (2021); Göçmen et al. (2022). Initial analysis revealed that combining the optimization of wake turbulence parameters with wake velocity parameters can result in the absence of a unique solution. This is in line with results found by Schreiber et al. (2020); Doekemeijer et al. (2022). Therefore the wake turbulence parameters are not optimized.

To achieve these goals, a novel optimization framework is developed, where the data is analyzed as a time series. Therefore, no prior binning is done based on environmental parameters, such as wind speed and wind direction. Binned analysis assumes balance and is valid when the magnitude and frequency of overestimation are in balance with the magnitude and frequency of underestimations. Otherwise, results can become skewed. Additionally, the volume of usable data becomes limited in binned observations, since even the downtime of a single turbine can introduce significant bias. Furthermore, the freeflow wind speed and wind direction, in addition to the wake parameters, are determined. This is crucial, since calibrating with inaccurate wind conditions can lead to results that misrepresent the true value of the wake model tuning parameters, adversely affecting the calibration results. While freeflow wind speed and direction are not the primary calibration targets, accurately determining these wind conditions ensures that results reflect the optimal value of the wake model tuning parameters.

To this end, a visual overview of the applied framework is provided in Figure 1. The framework is divided into six distinct segments which are described throughout the paper. First, section 2 gives a description of the case study wind farm and the input SCADA data. Different filtering procedures are described in order to ensure that no time windows are considered which are not representative for normal operation. Then, in section 3 a description of the wake model is given and a sensitivity study is performed on the velocity deficit parameters of the considered wake model. The optimization framework is described in section 4, followed with a validation of the acquired results based on energy ratio plots. Results are presented based on wind speed, wind direction, and their joint dependence. Furthermore, the Pearson correlation is analyzed for the optimized parameters.

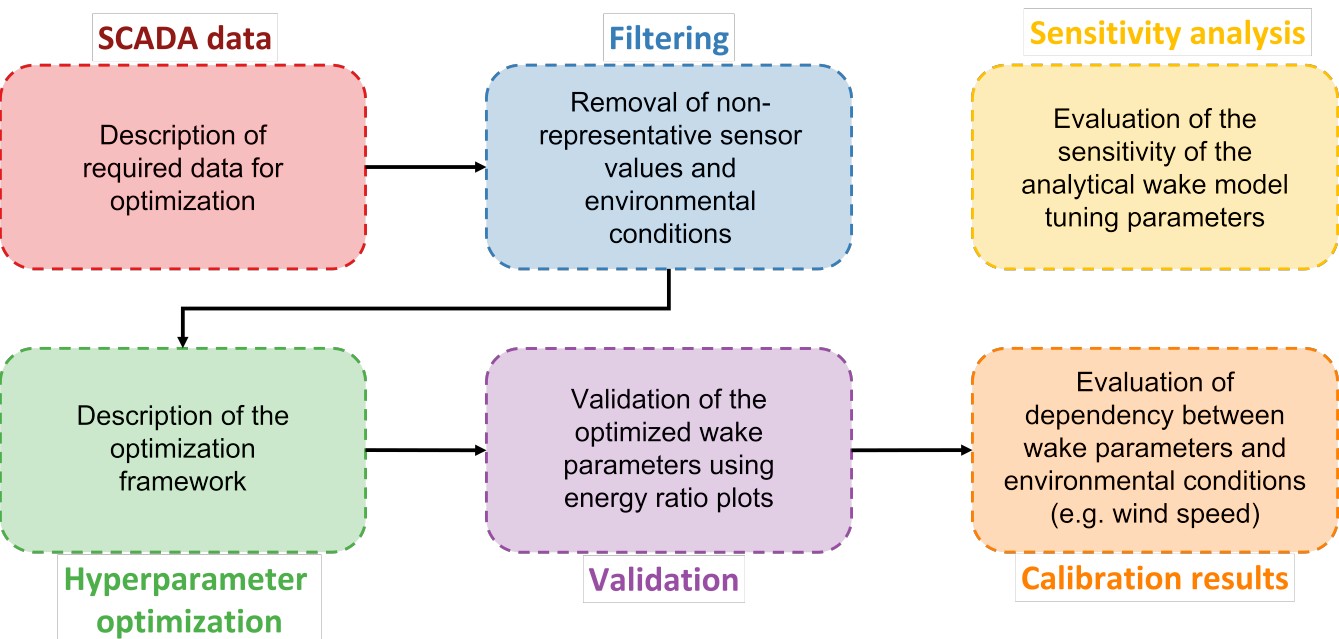

**Figure 1.** An illustrative overview of the applied framework is as follows: The SCADA data required, highlighted in red, and the filtering procedure, highlighted in blue, are described in section 2. The Sensitivity analysis, highlighted in yellow, is described and performed in section 3. Subsequently, the hyperparameter optimization, highlighted in green, the validation, highlighted in purple, and the results, highlighted in orange are all addressed in section 4.

## 2   Case study wind farm and the filtering and processing of SCADA data

In the following Section, the case study wind farm, together with the surrounding topography, is discussed. Then, the input data is described. This is followed by a description of the processing and filtering procedure applied to remove abnormal data. Finally, the effect of inter-farm interaction is identified and conditionally filtered out.

## 2.1 Available wind farm data for model calibration

The calibration in this paper is performed on one irregularly spaced offshore wind farm comprising large bottom-fixed wind turbines, each with a rated capacity of over 8.0 MW. The farm can produce over 300 MW of active power at full capacity. Figure 2 shows the surrounding topography to which the offshore wind farm is subjected. The performance of the offshore wind farm is affected by wakes of neighbouring wind farms between west-southwest and north east. The wind farm experiences coastal effects between east-northeast and east-southeast, due to its location along an irregular coastline with scarce low-rise structure. The coastline stretching from east-southeast to south-southwest is uniform, but features a combination of both low-rise and high-rise structures. The wind farm experiences wind from the sea from south-southwest to west-southwest and from northeast to east-northeast. In practice, there is a transition zone present between the above-mentioned zones, where the wind farm may be subject to a combination of inflow conditions from neighbouring wind farms, coastal effects or wind from the sea.

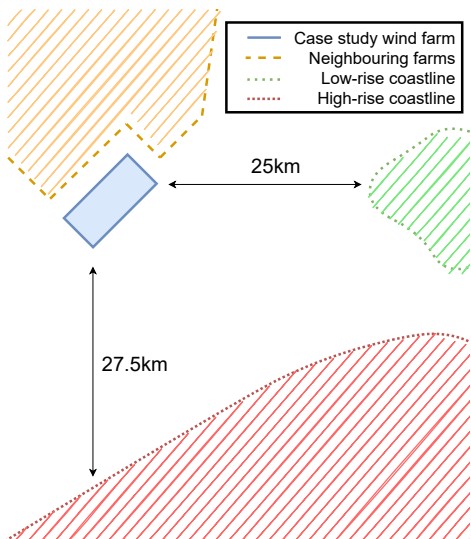

**Figure 2.** Illustration showing the surrounding topography impacting the offshore wind farm. The case study wind farm is visualized in the center of the figure with a blue rectangle, while in yellow the neighbouring wind farms are depicted. In green the low-rise coastline is illustrated and in red the high-rise coastline is outlined.

## 2.2 Input data and filtering of non-representative data windows

SCADA data over a four-year period is used to perform the presented analyses. Both the mean value and the variance of variables are calculated for the 10-minute intervals. Measurement channels include the active power, wind speed, wind direction, and nacelle position of wind turbines. It is essential to properly filter this SCADA data in order to not skew the calibration of the wake parameters due to windows of abnormal operation. This section explains the different steps that are applied within the paper in order to filter out these operating windows. These different conditions consist of underperformance, turbine downtime,

and alarm conditions, as acquired by the turbine control system.

Considered unrealistic sensor values are wind vane sensor or anemometer stuck faults. When the variance of the wind speed or wind direction for a 10-minute timestamp is effectively zero, it is assumed that the wind vane or anemometer is stuck, respectively. Similarly, a sensor stuck fault is assumed if consecutive 10-minute averages of the wind speed and wind direction acquired from the wind vane or anemometer are exactly equal.

Underperformance is subdivided into grid curtailment and turbine derating. The former refers to the state when the power set point of a wind farm is purposefully reduced below the maximum possible power output for a given environmental conditions. Specifically, grid curtailment is usually a contractually defined condition or requirement, used to mitigate overloading of the grid. Derating refers to the intentional decreasing of the rated power output of a wind turbine, with the aim to improve machine lifetime by reducing the mechanical loads on wind turbine components. Both types of under-performance are considered abnormal behaviour and are therefore filtered out. Daems et al. (2021, 2023) can be consulted for a further breakdown of the identification of these annotations. Additionally, power curve filtering, similar to the methods outlined by Doekemeijer et al. (2022), is performed, but with a stricter acceptance criteria around rated conditions, specifically adjusted to exclude grid curtailment near rated capacity.

During operating windows accompanied with low active power, the turbine is annotated as inactive. Low active power corresponds with low thrust loads and therefore has a limited effect on the internal wind farm flow field. The annotation is later used to remove the inactive turbines from the optimization for the given timestamp. When more than half of the wind farm is annotated as inactive, the data will not be considered for optimization.

Alarm annotations are acquired from status logs. The data is filtered out when the turbine status log occurs with high active power production. Results from the filtering procedure can be seen in Figure 3. In red the rejected SCADA data can be seen, while in green the accepted SCADA data is shown. Turbine inactivity is visualized by the blue colour.

## 2.3 Filtering of inter-farm effects

The effect of neighbouring farms on the optimized results is removed by filtering data based on wind direction. The effect of neighbouring farms is quantified by calculating the normalized absolute and relative power losses per wind speed and wind direction, as described by Equations 1, and 2, respectively. Here, $P_{int}$ represents the power production of the wind farm for a given wind speed and wind direction, as if only the wind farm itself is constructed in the concession zone. On the other hand, $P_{ext}$ represents the obtained power production for a given wind speed and wind direction considering the entire wind farm concession.

$$P_{loss,abs}(ws,wd) = \frac{P_{int}(ws,wd) - P_{ext}(ws,wd)}{\max_{ws,wd}\left(P_{int}(ws,wd) - P_{ext}(ws,wd)\right)} \tag{1}$$

$$P_{loss,rel}(ws,wd) = 100 \cdot \frac{P_{int}(ws,wd) - P_{ext}(ws,wd)}{\max_{wd} P_{int}(ws,wd)} \tag{2}$$

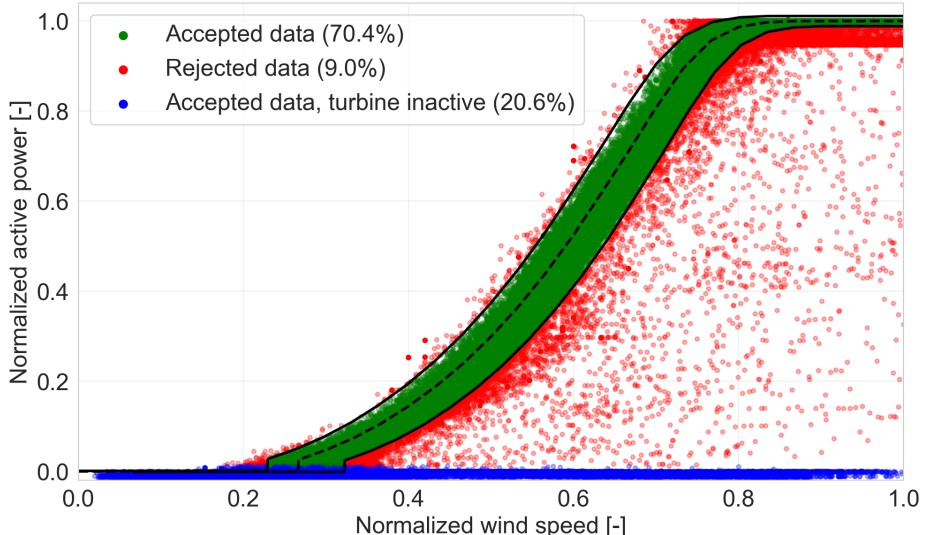

**Figure 3.** Comparison between 10-minute SCADA data before and after power-curve filtering.

Figure 4 and 5 display the normalized absolute and relative power losses per wind speed and wind direction, calculated using the TurbOPark wake model. A significant loss is observed between 250 and 50 degrees. Despite representing a large portion of the data, maintaining the uniformity of the freeflow inflow is critical for the accurate estimation of wake parameters. Measurements further reveal an increase in turbulence for the wind coming from neighbouring wind farms, as shown by the 10-minute turbulence intensity $TI$, variance in wind speed $\sigma_{ws}^2$, and variance in wind direction $\sigma_{wd}^2$ in Figure 6 and 7 and 8, respectively. Both heterogeneous inflow and increased turbulence affect the optimization process. This becomes evident when analyzing the resultant accumulated absolute and relative errors between the results acquired from FLORIS and the SCADA data, as shown in Figure 9 and 10. The error metrics, denoted as $\epsilon_{abs}$ for the accumulated absolute error and $\epsilon_{rel}$ for the accumulated relative error, are defined by Equation 3 and 4, respectively. $N_T$ represents the number of active turbines, while $\bar{P}_i$ and $\hat{P}_i$ are active power from SCADA data and calculated power from the wake model for turbine $i$, respectively. A noticeable increase in the error can be observed where the external wake losses are the highest. Therefore, in order to obtain results that accurately reflect unaffected freeflow conditions, it is necessary to filter out the inter-farm effects from the data.

$$\epsilon_{abs} = \sum_{i=1}^{N_T} |\bar{P}_i - \hat{P}_i| \tag{3}$$

$$\epsilon_{rel} = \frac{\sum_{i=1}^{N_T} |\bar{P}_i - \hat{P}_i|}{\sum_{i=1}^{N_T} \bar{P}_i} \tag{4}$$

As such, data between 250 and 50 degrees are excluded from the wind speed-dependent results. This further underlines the importance of carrying out a simulation of the entire wind farm cluster for obtaining precise wake parameters, especially

when the wind direction overlaps with neighbouring wind farms. This presents its own set of complications, such as fitting of thrust-and-power curves and having limited knowledge on the operational status of the neighbouring turbines, among others. Moreover, an increase in the relative power error between the SCADA data and the analytical wake model results can be observed for wind originating from the irregular coastline. This suggests a potential high inflow heterogeneity, which the homogeneous inflow assumption fails to account for.

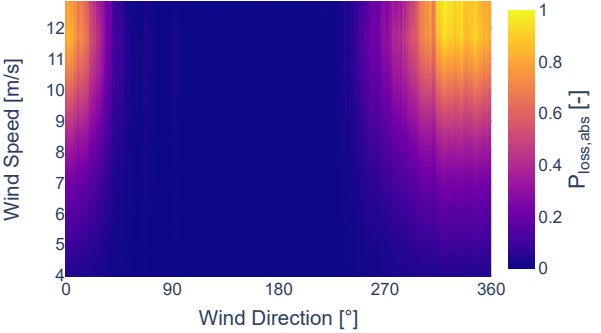

**Figure 4.** Normalized absolute power losses [-] due to external wake effects, calculated using the TurbOPark model.

**Figure 5.** Normalized relative power losses [%] due to external wake effects, calculated using the TurbOPark model.

## 3  Wake model and input parameter sensitivity

In the following Section the considered wake model used for the optimization framework is described. To identify the importance of the tuning parameters of the wake model, a sensitivity study is performed on the wake model.

### 3.1  Wake Model Description

The Gauss-Curl Hybrid model, commonly referred to as the GCH model, is the wake model used in this work and has been briefly mentioned in subsection 1.1. The decision to calibrate the GCH model is influenced by its widespread use in relevant literature (Bastankhah and Porté-Agel (2016); Archer et al. (2018); Fleming et al. (2019, 2020, 2021); Hamilton et al. (2020); Doekemeijer et al. (2021); Simley et al. (2021); van Beek et al. (2021); Doekemeijer et al. (2022); Göçmen et al. (2022)), which provides a strong foundation for comparative analysis. The wake velocity model of the GCH wake model can be subdivided into two areas: The near-wake region and the far-wake region. Within the near-wake region the wake model is modeled as a linearly converging cone. Assuming no misalignment, the width of the cone is equal to the rotor diameter at the turbine hub and becomes zero when the near-wake region ends. The start of the far-wake region, denoted as $x_0$, is characterized by a two-dimensional Gaussian distribution. The transition from the near-wake to the far-wake is governed by Equation 5. Here

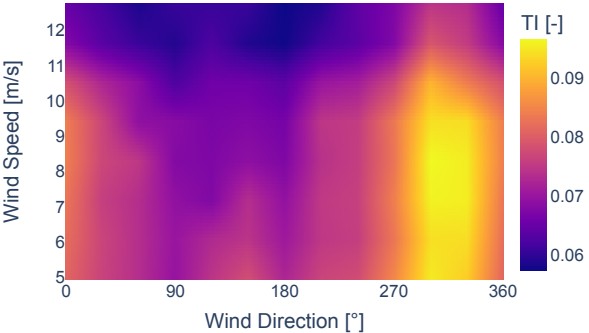

**Figure 6.** Freeflow turbulence intensity as function of wind speed and wind direction.

**Figure 7.** Freeflow wind direction variance as function of wind speed and wind direction.

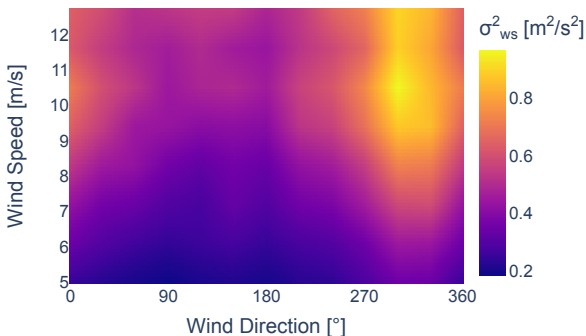

**Figure 8.** Freeflow wind speed variance as function of wind speed and wind direction.

$D_{rotor}$ represents the rotor diameter of the wind turbine, $C_T$ is the thrust coefficient of the turbine, $I_{rotor}$ stands for the turbine specific turbulence intensity, and $\alpha$ and $\beta$ are the tuning parameters, referenced earlier.

$$x_0 = \frac{D_{rotor}\left(1 + \sqrt{1 - C_T}\right)}{\sqrt{2}\left(4\alpha I_{rotor} + 2\beta\left(1 - \sqrt{1 - C_T}\right)\right)} \tag{5}$$

Without yaw misalignment, the far-wake profile can be described using Equation 6. Here $U_\infty$ represents the upstream wind speed, while $U$ is the wind speed within the 3D Euclidean space $(x,y,z)$, with its origin at the turbine hub. The $x$ coordinate aligns with the wind direction, whereas $y$ and $z$ stand perpendicular to the wind direction. The $z$ coordinate is defined as positive in the upward direction. Additionally, $\sigma_y$ and $\sigma_z$ represent the standard deviation of the Gaussian distribution in the $y$ and $z$ direction, respectively.

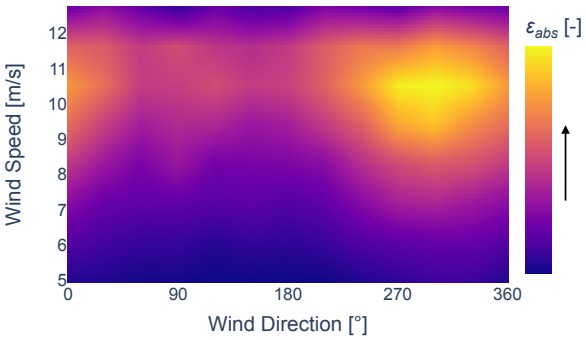

**Figure 9.** Normalized power error per wind speed and wind direction bin for the GCH model.

**Figure 10.** Normalized relative power error per wind speed and wind direction bin for the GCH model.

$$\frac{U(x, y, z)}{U_\infty} = 1 - \left(1 - \sqrt{1 - \frac{\sigma_{y,0}\sigma_{z,0}}{\sigma_y \sigma_z} C_T}\right) \exp\left(-\left(\frac{y^2}{2\sigma_y^2} + \frac{z^2}{2\sigma_z^2}\right)\right) \tag{6}$$

The progression of the standard deviation of the Gaussian distribution can be described by Equation 7 and Equation 8. Here, Equation 9 defines the standard deviation of the Gaussian at $x_0$. The wake expansion coefficients $k_y$ and $k_z$ are described as function of the specific turbulence intensity and the tuning parameters $k_a$ and $k_b$, according to Equation 10.

$$\sigma_y = \sigma_{y,0} + (x - x_0) k_y \tag{7}$$

$$\sigma_z = \sigma_{z,0} + (x - x_0) k_z \tag{8}$$

$$\sigma_{y,0} = \sigma_{z,0} = \frac{D_{rotor}}{2\sqrt{2}} \tag{9}$$

$$k_y = k_z = k_a I_{rotor} + k_b \tag{10}$$

In this work, the set of tuning parameters within the velocity deficit model of the GCH wake model, $\mathbf{\Omega} = \{k_a, k_b, \alpha, \beta\}$, are considered for optimization. Model specifications, including the velocity, deflection, turbulence, and combination model, can be found in Table 1. In addition, Table 1 also provides information on the atmospheric parameters.

The reference values for the parameters are the reference values that are used within the FLORIS framework, while the minimum and maximum values, defined in Table 2, are acquired from Doekemeijer et al. (2020); van Beek et al. (2021).

### 3.2 Sensitivity Analysis

In order to assess the sensitivity of the tuning parameters, the total-order Sobol indices are computed by performing Sobol' method for sensitivity analysis with Saltelli's extension. The Sobol' method, as described in Sobol (2001), is a variance-

| | GCH | References |
|---|---|---|
| Wake Velocity Model | Gauss | Bastankhah and Porté-Agel (2014); Niayifar and Porté-Agel (2015) |
| Wake Deflection Model | Gauss | Bastankhah and Porté-Agel (2014); King et al. (2021) |
| Wake Turbulence Model | Crespo Hernandez | Crespo and Hernandez (1996) |
| Wake Combination Model | SOSFS | Katić et al. (1987) |
| Air Density | 1.225 | |
| Turbulence Intensity | 0.06 | |
| Wind Shear | 0.12 | Gebraad et al. (2016) |
| Wind Veer | 0.0 | Gebraad et al. (2016) |

**Table 1.** Overview of considered submodels and atmospheric parameters.

| Parameter | Physical representation | Min: $\Omega^{min}$ | Max: $\Omega^{max}$ | Reference value: $\Omega^{ref}$ |
|---|---|---|---|---|
| $k_a$ | wake expansion | 0.05 | 1.5 | 0.38 |
| $k_b$ | wake expansion | 0.0 | 0.02 | 0.004 |
| $\alpha$ | near-wake to far-wake transition | 0.125 | 2.5 | 0.58 |
| $\beta$ | near-wake to far-wake transition | 0.015 | 0.3 | 0.077 |

**Table 2.** Parameter space, $\widetilde{\Omega}$, considered for the sensitivity study and the optimization of the Gauss-Curl hybrid wake model.

based global sensitivity analysis tool that quantifies the degree of contribution of each individual input parameter to the output variance. The method involves variance decomposition of the model output to input variations, which defines the magnitude of the Sobol indices, meaning that the magnitude of the Sobol index is directly proportional to the sensitivity of the input parameter on the model output. The method is capable of generating both first-order indices, which ignores interactions between

315 input variables, and total-order indices, which consider both the contribution of input variations on the output variance and input interactions. To ensure that the full parameter space is covered, an evenly distributed quasi-random and low-discrepancy sequence is required. This sequence is created by performing Saltelli's extension by Saltelli (2002); Saltelli et al. (2010) on the Sobol' sequence (Sobol (2001)). Then, Sobol' method is performed on wind speeds varying from $5\,m/s$ to $12\,m/s$ in $0.5\,m/s$ increments and all wind directions at $12°$ intervals.

The total-order Sobol indices, $S_T$, for the parameters $k_a$, $k_b$, $\alpha$ and $\beta$ are depicted in Figure 11, 12, 13 and 14, respectively. The results reveal that the sensitivity of the GCH model is primarily governed by the parameter $k_a$. It is also observed that the sensitivity of the parameter $\alpha$ increases with higher wind speeds from two anti-parallel directions. Generally, a high degree of symmetry can be observed.

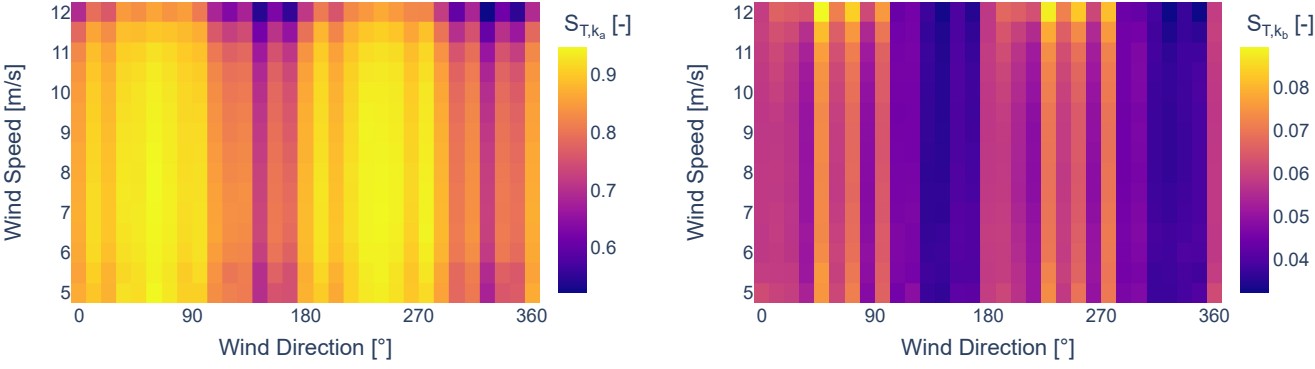

**Figure 11.** Total sobol indices for $k_a$.

**Figure 12.** Total sobol indices for $k_b$.

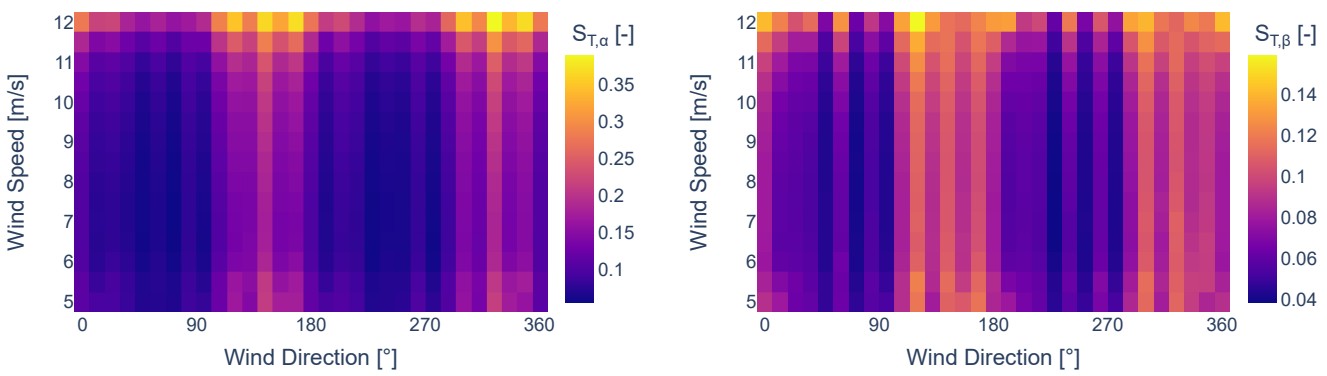

**Figure 13.** Total sobol indices for $\alpha$.

**Figure 14.** Total sobol indices for $\beta$.

## 4 Optimization using SCADA Data

This section discusses the optimization framework performed on the GCH model and validates these results using energy ratio
plots. Then, the optimized parameters are presented as a function of wind speed and wind direction.

### 4.1 Optimization Framework

A hyperparameter optimization framework, named Optuna (Akiba et al. (2019)), is used to optimize the parameters of the
velocity deficit model. Optuna serves as a specialized framework for hyperparameter optimization, aimed at finding the ideal
set of parameters for machine learning such as the learning rates and number of hidden layers in a neural network or the depth
in a decision tree.

In order to reduce the optimization time and number of local minima, the optimization process is subdivided into three stages:

– Stage 1: The first stage involves the optimization of the wind speed.

 – Stage 2: The second stage optimizes both the wind speed and wind direction.

– Stage 3: In the final stage, wind speed, wind direction and wake parameters are optimized together.

In the first and second stages, the Quasi Monte Carlo (QMC) Sampler by Bergstra and Bengio (2012) is used to explore the entire tuning parameter search space. QMC sequences, known for their lower discrepancies compared to standard random sequences, are effective in initially exploring the search space more efficiently than the Tree-Parzen Estimator (TPE) algorithm
 (Bergstra et al. (2012)). At these stages, obtaining a precise estimate is less critical and instead a general approximation is preferred. For the final stage the multivariate Tree-structured Parzen Estimator (TPE) algorithm is used as sampling algorithm and is combined with 50 random start-up trials. The TPE algorithm fits a Gaussian Mixture Model (GMM) to a set of parameter values linked to the best objective values. Concurrently it creates a separate GMM pertaining to the rest of the parameter values. The TPE algorithm then chooses the parameter set that maximizes the ratio between the GMM associated with the best objective
 values and the set of remaining parameters. For further information on the framework and the algorithm, Akiba et al. (2019) can be conducted.

The cost function used in these stages consists of two components, $f$ and $g$,

$$f\left(U, \phi, \boldsymbol{\Omega}\right) = \frac{1}{N_T} \sum_{i=1}^{N_T} \left(\bar{P}_i - \widehat{P}_i\left(U, \phi, \boldsymbol{\Omega}\right)\right)^2 \tag{11}$$

$$g\left(U, \phi, \boldsymbol{\Omega}\right) = \left(\sum_{i=1}^{N_T} \bar{P}_i - \sum_{i=1}^{N_T} \widehat{P}_i\left(U, \phi, \boldsymbol{\Omega}\right)\right)^2, \tag{12}$$

 where $\bar{P}_i$ is the active power from the SCADA data of turbine $i$, $\hat{P}_i$ is the power of turbine $i$ derived from FLORIS, $U$ is the freeflow wind speed, $\phi$ is the freeflow wind direction, and $\boldsymbol{\Omega}$ is the set of wake velocity deficit parameters. Function $f$ is divided by the number of active turbines, $N_T$, for normalization purposes. Intuitively, $f$ is a cost function that penalizes large errors on turbine-level, while $g$ is a cost function that penalizes large errors on farm-level.

The weight between the two components, $f$ and $g$, is defined by the normalized constants $a$ and $b$. A preliminary optimization
 of $a$ and $b$ is conducted to find the combination that results in the fastest convergence rate and smallest normalized squared error per wind turbine. The criteria is that $a + b = 1$, and $a > 0$, $b > 0$. The identified combination that has been determined is then used for the optimization of the wind speed, wind direction, and wake velocity parameters. Given the importance of penalizing large errors on turbine-level, a final ratio $\frac{a}{b}$ of 4.0 is considered, emphasizing the importance of function $f$ over $g$.

For the first optimization step, the freeflow wind speed $\mathcal{U}_1$ is optimized given a constant value for the initial freeflow wind
 direction $\bar{\phi}_\infty$ and wake parameters $\boldsymbol{\Omega}^{ref}$. This is done by minimizing the cost function shown in Equation 13. The initial

freeflow wind speed $\bar{U}_\infty$ is determined as the mean wind speed of the first turbine row of the wind farm at the upstream edge. The initial freeflow wind direction is determined from the median of all wind turbines within the wind farm. The algorithm is allowed to shift the wind speed by up to 40%, as relatively large inconsistencies are observed between the initially determined freeflow wind speed and the active power of the wind farm. Furthermore, it is also constrained that the optimized freeflow wind speed is larger than 4.0 $m/s$. The aim for this first optimization stage is to match the freeflow wind speed better with the FLORIS simulation, given the power curve and the individual turbine powers are known.

$$\text{minimize} \quad a \cdot f\left(\mathcal{U}_1, \bar{\phi}_\infty, \mathbf{\Omega}^{ref}\right) + b \cdot g\left(\mathcal{U}_1, \bar{\phi}_\infty, \mathbf{\Omega}^{ref}\right) \tag{13}$$

$$\text{subject to} \quad 0.6\bar{U}_\infty \leq \mathcal{U}_1 \leq 1.4\bar{U}_\infty \tag{14}$$

In the second optimization stage, both the freeflow wind speed $\mathcal{U}_2$ and wind direction $\Phi_2$ are optimized. The optimization of the latter is important in order to minimize the biases of the directional wind vanes. To this end, a search space of 15 degrees from the initial guess $\bar{\phi}_\infty$ is considered. It is opted to allow a smaller optimization range for the freeflow wind speed, as it is assumed that this variable has already been optimized to the value $\mathcal{U}_1^*$ in the previous step. The initial value imposed to the optimization problem $\mathcal{U}_2$ is equal to the previous estimate $\mathcal{U}_1$. This variable is allowed to be adjusted by up to 5%. This can be seen in Equation 15.

$$\text{minimize} \quad a \cdot f\left(\mathcal{U}_2, \Phi_2, \mathbf{\Omega}^{ref}\right) + b \cdot g\left(\mathcal{U}_2, \Phi_2, \mathbf{\Omega}^{ref}\right) \tag{15}$$

$$\text{subject to} \quad 0.95\mathcal{U}_1^* \leq \mathcal{U}_2 \leq 1.05\mathcal{U}_1^* \tag{16}$$

$$\text{and} \quad \bar{\phi}_\infty - 15 \leq \Phi_2 \leq \bar{\phi}_\infty + 15 \tag{17}$$

In the final step, the three design parameters are optimized simultaneously. These consist of the freeflow wind speed $\mathcal{U}_3$, the freeflow wind direction $\Phi_3$, and the wake parameters set $\mathbf{\Omega} = \{k_a, k_b, \alpha, \beta\}$. The initial value of $\mathcal{U}_3$ and $\Phi_3$ are imposed as the average of the values obtained after the two previous optimization steps, respectively, $\mathcal{U}_1^*$ and $\mathcal{U}_2^*$, and $\bar{\phi}_\infty$ and $\Phi_2^*$. In addition, the wake parameters are varied. The minimum, initial, and maximum values of the parameter set $\mathbf{\Omega}$ can be found in Table 2, defining the parameter space $\tilde{\mathbf{\Omega}}$.

$$\text{minimize} \quad a \cdot f\left(\mathcal{U}_3, \Phi_3, \mathbf{\Omega}\right) + b \cdot g\left(\mathcal{U}_3, \Phi_3, \mathbf{\Omega}\right) \tag{18}$$

$$\text{subject to} \quad 0.95\mathcal{U}_2^* \leq \mathcal{U}_3 \leq 1.05\mathcal{U}_2^* \tag{19}$$

$$\text{and} \quad \Phi_2^* - 15 \leq \Phi_3 \leq \Phi_2^* + 15 \tag{20}$$

$$\text{and} \quad \mathbf{\Omega} \in \tilde{\mathbf{\Omega}} \tag{21}$$

## 4.2 Validation of Results

Results are validated by comparing the accumulated relative wake model error between the calibrated model and the reference model for wind directions between 50 and 250 degrees. The error metric used is comparable to the one described in Nygaard et al. (2022). However, the described error metric focuses on the accumulated error between SCADA data and the wake model for each wind turbine, described by Equation 4.

Figure 15 shows the accumulated relative error without calibration of the tuning parameters in blue and after calibration of the tuning parameters in red. Here, it is evident from the figure that the optimization effectively reduces the accumulated error. Before calibration, the median error is 15.7%, with interquartile errors from 12.4% to 20.5%. After calibration, the median error decreases to 14.2%, with interquartile errors from 11.4% to 18.4%. This represents an improvement of 9.3% for the median value and relative improvements of 8.4% and 10.2% for the interquartile range.

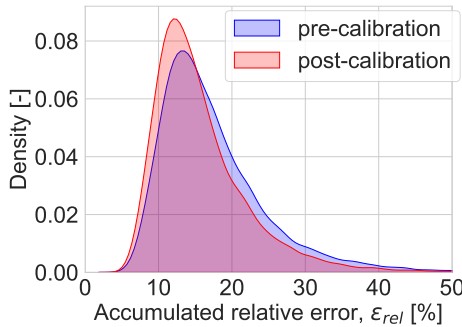

**Figure 15.** Accumulated relative error between SCADA data and the GCH wake model on turbine level for time series data and wind directions ranging from 50 to 250 degrees. The reference model is shown in blue, while the calibrated model is shown in red.

Additionally, a comparison of the energy ratios is performed for clustered wind turbines within the wind farm, similar to the energy ratio defined by Doekemeijer et al. (2022). The energy ratios are computed for two groups of wind turbines, as shown in Figure 16. Here group 1 represents the turbines from Figure 17 and group 2 represents the turbines from Figure 18. Figure 17 shows the energy ratios for a group of clustered wind turbines far from the neighbouring wind farms, while Figure 18 shows the energy ratios for a group of clustered wind turbines close to neighbouring wind farms. Here, SCADA data is compared to the simulation results acquired from the FLORIS framework. For a given bin width, the energy ratios are presented, specifically for this case at 9 and 10 $m/s$. The line represents the median for a bin width of 3 degrees, while the opaque region signifies the area between the $25^{th}$ and $75^{th}$ percentile marks. The bottom plot displays the amount of data points used to determine the median and percentile indicators. When the energy ratios obtained from FLORIS align with the energy ratios derived from SCADA data, it can be concluded that the filtering procedure is effective in removing transient data. Moreover, this signifies that the optimization process has reached convergence.

Both Figure 17 and 18 show good agreement between the energy ratios obtained using FLORIS and the energy ratios derived from SCADA data when the inflow wind is not affected by neighbouring wind farms. Similarly, both Figures show an increase

in the mean error for wind coming from neighbouring wind farms. The magnitude of the observed error differs significantly between Figure 17 and 18. This can be attributed to the proximity of the wind turbines from Figure 18 to neighboring wind farms, which is closer compared to the wind turbines from Figure 17. The neighbouring wind farms create both heterogeneous inflow and a wake where homogeneous unaffected inflow is assumed by the model. The error will be most present close to the neighbouring wind farms since the heterogeneous effect of wake propagation recedes due to wake recovery further from

the neighbouring wind farms. The wind turbines from Figure 17 will therefore experience less heterogeneous inflow from neighbouring wind farms than the wind turbines from Figure 18. The increase in the error agrees with the findings from Figure 9 and 10, where a larger final optimization bias is observed for wind coming from neighbouring wind farms.

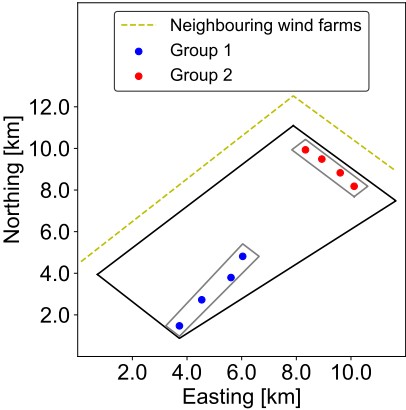

**Figure 16.** Overview of the groups of wind turbines used for energy ratio analysis. Group 1 represents a group of wind turbines far from neighbouring wind farms, while group 2 represents a group of wind turbines close to neighbouring wind farms.

### 4.3 Optimization Results

The set of optimized parameters of the GCH velocity deficit, $\mathbf{\Omega}^* = \{k_a, k_b, \alpha, \beta\}$, are optimized based on the cost function, as

discussed in subsection 4.1. The Pearson correlation matrix for the set of parameters can be seen in Figure 19. All variables exhibit weak correlations with each other, as indicated by all correlation coefficients being below 0.2. This suggest that none of the variables share a strong linear relationship.

The expected value over wind speed and wind direction can be seen in Figure 20 and 21, respectively. The line indicates the median value per wind speed or wind direction bin, while the opaque area indicates the area between the $25^{th}$ and $75^{th}$

percentile marks. The dotted black line represents the reference value within the FLORIS framework. The bottom plot displays the amount of data that is used to determine the median and percentile indicators. Figure 20 shows the expected wind speed based on wind directions between 50 and 250 degrees. This filtering is performed to remove the effect of external wakes on the parameter estimation, as discussed in subsection 2.3.

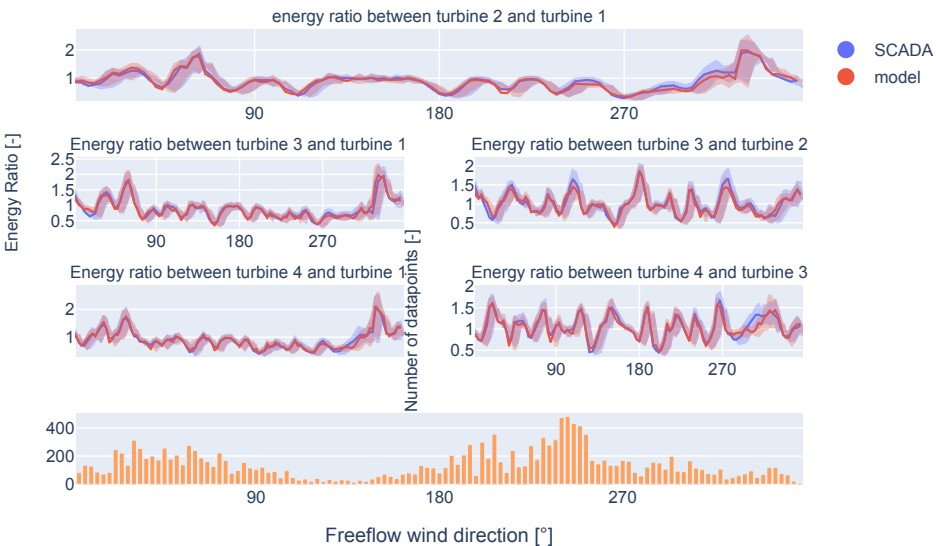

**Figure 17.** Validation of the optimized results for group 1 using the energy ratio for the GCH wake model for wind speeds between 9 and 10 $m/s$. These turbines are located in far proximity from neighbouring wind farms.

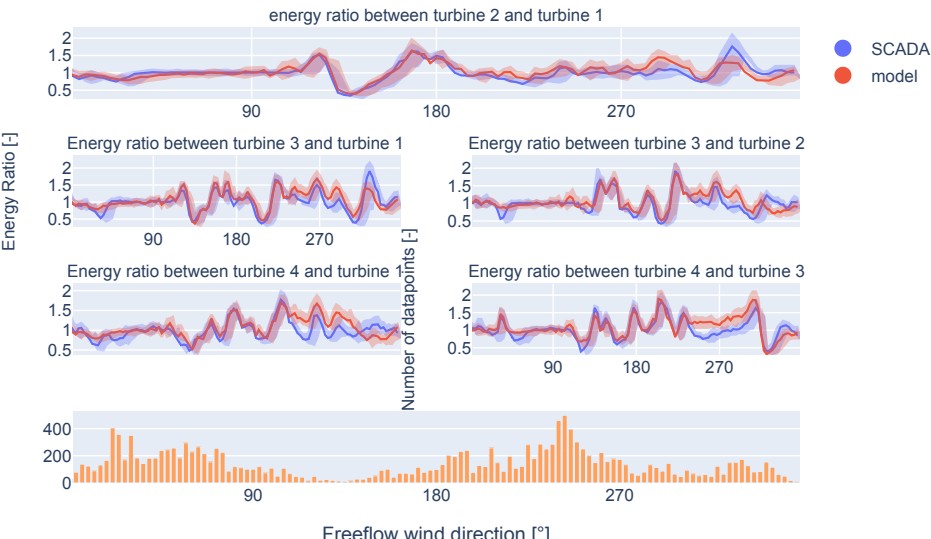

**Figure 18.** Validation of the optimized results for group 2 using the energy ratio for the GCH wake model for wind speeds between 9 and 10 $m/s$. These turbines are located in close proximity from neighbouring wind farms.

Parameter $k_a$, recognized as the most sensitive parameter in the GCH model from the sensitivity study, converges to a value
below its reference value, whereas the remaining parameters converge to above their respective reference value. Given the

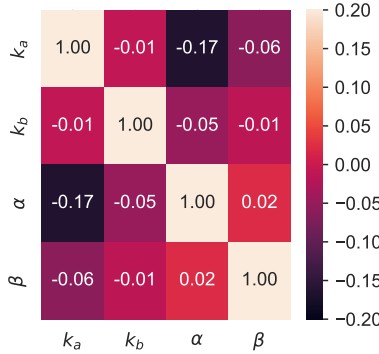

**Figure 19.** Pearson correlation matrix for the velocity deficit parameters of the GCH wake model.

direct effect of $k_a$ on wake recovery, as described by Equation 6-10, this implies that the GCH wake model underestimates the impact of internal wakes when baseline parameters are used in combination with a reference turbulence intensity of 0.06. This is further confirmed by calculating wake losses for the wind conditions specific to the local site using the optimized tuning parameters and comparing these to the losses obtained with the reference parameters. This comparison shows a relative increase in wake losses of 14% compared to the losses obtained with the reference parameters. The trend observed for $\alpha$ agrees with the findings from the sensitivity analysis. As the importance of $\alpha$ increases, its value shows a consistent convergence and decrease in variance. The same is observed for the parameter $k_a$, where its variance increases as its importance decreases. The wide spread observed for parameters $k_b$ and $\beta$ is in line with the expectations derived from the sensitivity analysis. A large variance is expected when non-sensitive parameters are optimized, since it should theoretically yield a uniform distribution in a Bayesian optimization scenario. The results do not perfectly align with a uniform distribution, implying the parameters are not fully non-sensitive.

The expected results for different wind directions are depicted in Figure 21, similar to how Figure 20 represents the parameters for different wind speeds. It is noticeable that $k_a$ shows nonlinearity with respect to the wind direction. A number of factors can contribute to this, including the layout of the wind farm, terrain features, data scarcity, or the different distributions (e.g. Weibull scaling parameters) per wind direction. To get a more comprehensive understanding of the joint relationship between wind speed and wind direction, the joint relationship between wind speed and wind direction for each parameter is analyzed.

The joint distribution of the wake velocity deficit parameters ($k_a$, $k_b$, $\alpha$, $\beta$) of the GCH wake model are illustrated in Figure 22, 23, 24 and 25, respectively. An increase in the values for $k_a$ and $k_b$ is visible between 250 and 50 degrees, which can be linked to neighbouring wind farms. Similarly, as identified in Figure 21, values for $k_a$ and $k_b$ are visibly higher from east to south, which is the direction closest to the coast. The increase can be contributed to an increase in turbulence from land or a wind speed gradient due to different topographic properties between land and sea.

As for parameters $\alpha$ and $\beta$, no distinct directional trends influenced by the layout or the placement of the wake are visible. However, a noticeable decrease of $\alpha$ and $\beta$ can be observed as wind speeds approach rated wind speed. It is important to note

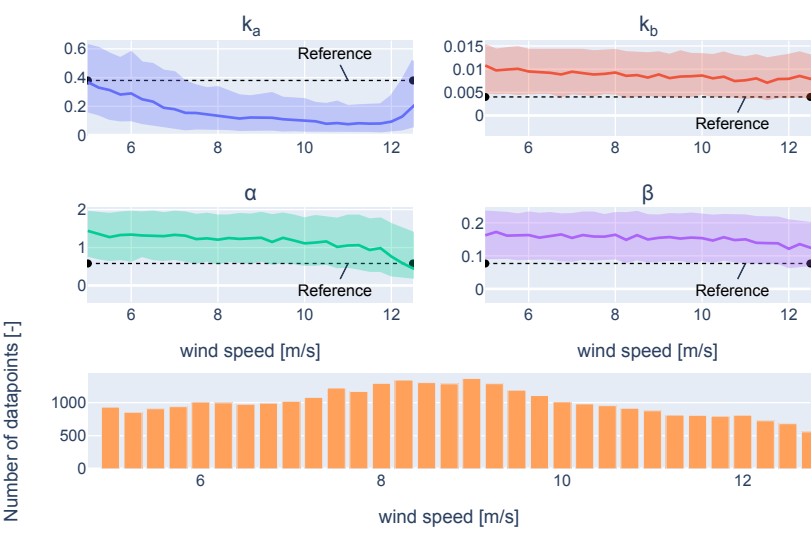

**Figure 20.** Expected value of the set of parameters, $\Omega^* = \{k_a, k_b, \alpha, \beta\}$, as function of wind speed.

that this study does not include wind speeds exceeding the rated value. Therefore any conclusions about the behaviour of these
parameters beyond this point cannot be made.

The difference between the identified tuning parameters and those presented in Niayifar and Porté-Agel (2016) and Trabucchi et al. (2017) can be partially explained by the constant turbulence intensity assumption, set to 0.06 for this study, since accurately determining the turbulence intensity based on SCADA data is not trivial. In reality, the turbulence intensity exhibits a strong dependency on wind speed and may also vary with wind direction. This explains the observed downward trend for the parameter $k_a$. Consequently, changing the ambient turbulence intensity requires additional calibration. Additionally, the metrics used in the optimized cost-function consider the collective power production of the wind farm, in contrast with the results from Niayifar and Porté-Agel (2016) and Trabucchi et al. (2017), which are based on the wake of a single wind turbine, specifically focusing on a set number of rotor diameters behind the wind turbine. Optimization at the scale of an entire wind farm requires the tuning parameters to account for flow physics that are inherently different from those encountered in a single wake case.

## 5 Conclusions

A reliable method for calibrating analytical wake models for both yield assessment and control purposes has been established in this study. Rather than utilizing binned data, this optimization process employs time series data. The calibration process

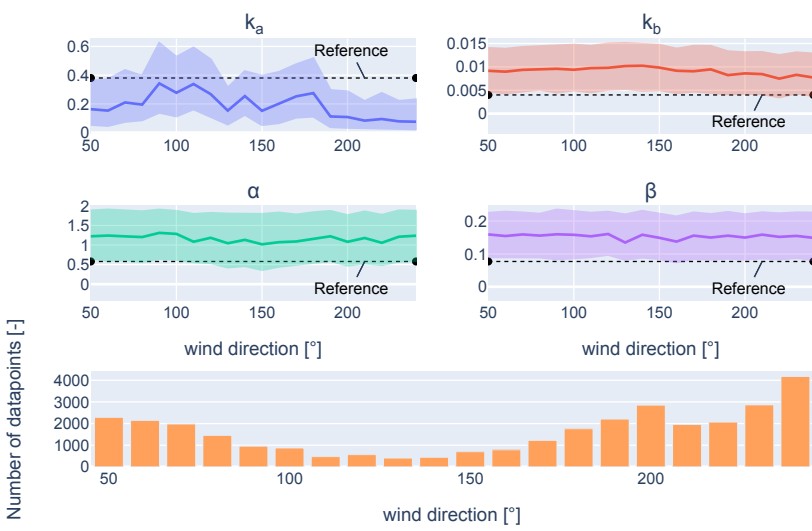

**Figure 21.** Expected value of the set of parameters, $\Omega^* = \{k_a, k_b, \alpha, \beta\}$, as function of wind direction.

is executed in three stages utilizing a Tree-Structured Parzen Estimator for optimization. The first two stages determine the freeflow wind speed and wind direction. These are based on a cost function that minimizes the error between the active power from the Supervisory Control and Data Data Acquisition (SCADA) system and the power output estimated by the FLOw Redirection and Induction in Steady State (FLORIS) framework. This work focuses on optimizing the wake velocity deficit parameters of the Gauss-Curl Hybrid wake model. However, the set of parameters, referred to as $\boldsymbol{\Omega}$, can easily be adjusted for different submodels or wake models.

A sensitivity analysis on the considered set of parameters reveals that parameter $k_a$ is highly sensitive, while parameters $k_b$ and $\beta$ display minimal sensitivity. It is further observed that the sensitivity of parameter $\alpha$ increases with wind speed, which jointly determines the near-wake region transition point with $\beta$, as wind speeds increase. This conclusion is reinforced by a decrease in the variance and expected value for $\alpha$ approaching rated wind speed. The substantial importance of one parameter over the others suggests that the model is subject to overparameterization. The effect of overparameterization can be further observed when analyzing the expected results for the set of parameters. The parameters with high sensitivity converge strongly, whereas parameters with low sensitivity retain considerable variance within the defined optimization boundaries. Comparing the optimized parameters to the baseline reveals that the baseline parameters underestimate the wake effects, which subsequently leads to an overestimation of the expected yield.

A significant increase in the resultant cost function error is observed for wind coming from neighbouring wind farms, due to the heterogeneous inflow and wake-added turbulence coming from these wind farms. This increase is also clear when analyzing

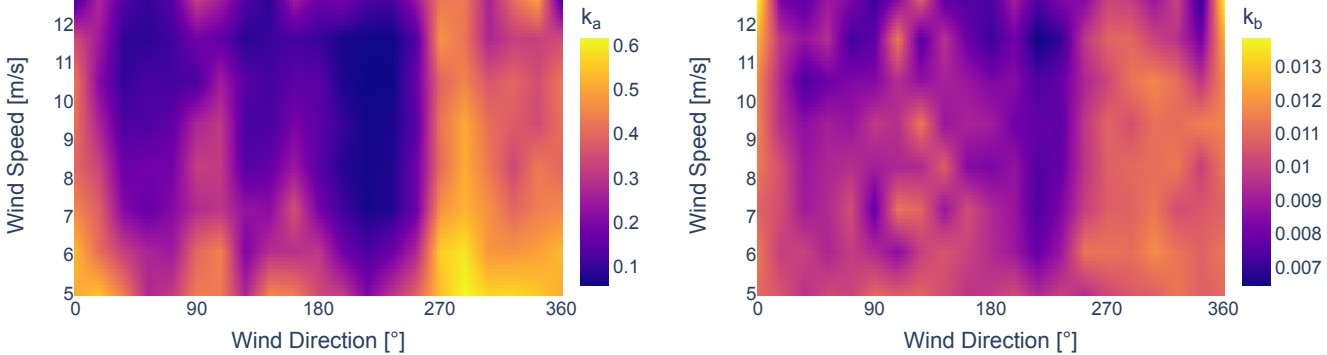

**Figure 22.** Joint distribution of the expected value for $k_a$ after calibration on SCADA data.

**Figure 23.** Joint distribution of the expected value for $k_b$ after calibration on SCADA data.

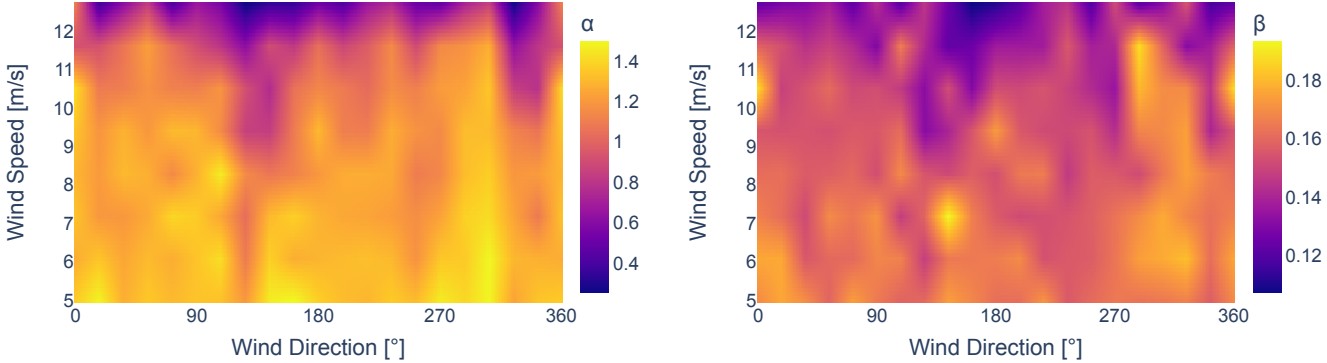

**Figure 24.** Joint distribution of the expected value for $\alpha$ after calibration on SCADA data.

**Figure 25.** Joint distribution of the expected value for $\beta$ after calibration on SCADA data.

the energy ratios between turbines near the neighbouring wind farms. Moreover, the resultant parameter set shows considerable variation between wind directions free from external wake effects and those impacted by neighbouring wind farms. The impact of the irregular coastline with low-rise buildings is also noticeable in the resultant cost function error. Additionally, the entire coastline, consisting of both low-rise and high-rise buildings, impacts the expected values for the wake expansion parameters,

$k_a$ and $k_b$.

Caution is advised when using these results with a turbulence intensity different from the reference value within the model. Additionally, it is important to acknowledge that these findings are site-specific and may not be directly transferable to other locations without careful consideration of the site-specific characteristics and recalibration. Future studies will involve a com-

parison of other calibrated analytical wake models with the Gauss-Curl Hybrid model. The substantial error in the cost function observed for wind coming from neighbouring wind farms suggest the necessity of including these wind farms during the optimization of wake parameters. Additionally, accounting for the wake blockage effect, particularly relevant in large wind turbine clusters, should then be considered. Future analysis will incorporate the turbine yaw misalignment, taking into account its uncertainties and impact on the resultant cost function error. Furthermore, together with a varying turbulence intensity, results will be analyzed under different atmospheric stabilities, and analysis of both diurnal and annual cycles will be conducted. Lastly, in this study the cost-function is based solely on the error associated with active power. For future research, it would be valuable to conduct a comparative analysis between the cost-function based on active power and one that incorporates the wind speed acquired from SCADA. Exploring the differences between using active power and wind speed in the cost-function could provide insights into their relative accuracy, and substantiate the choice of one metric over the other.

*Code and data availability.* The code used in this study is available from the FLORIS repository by NREL and the Optuna repository. For those interested in the Python implementation of the methodology, please reach out to the corresponding author. The SCADA data used within this work is not openly available due to confidentiality constraints.

*Author contributions.* DvB was responsible for the conceptualization, data curation, formal analysis, investigation, methodology, project administration, coding of software, validation, visualization and writing of the original draft. PJD and TV were responsible for conceptualization, methodology, supervision, paper reviewing, and editing. JH and ARN were responsible for conceptualization, funding acquisition, methodology, supervision, final phase paper reviewing, and final phase editing.

*Competing interests.* ARN is member of the editorial board of the Wind Energy Science journal. The peer-review process was guided by an independent editor, and the authors have also no other competing interests to declare.

*Acknowledgements.* The authors would like to acknowledge the support of De Blauwe Cluster through the project Cloud4Wake. Furthermore, the authors acknowledge the support via theMaDurOS program from VLAIO (Flemish Agency for Innovation and Entrepreneurship) and SIM (Strategic Initiative Materials) through project SBO SeaFD. The authors would also like to acknowledge the energy transition fund for their support of the Poseidon project. This research was supported by funding from the Flemish Government under the "Onderzoeksprogramma Artificiële Intelligentie (AI) Vlaanderen" programme.

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
