# Peer review of "Hyperparameter tuning framework for calibrating analytical wake models using SCADA data of an offshore wind farm applied on FLORIS"

_Wind Energy Science, 2023_

## Referee Comment (RC1)

**Review for WES-2023-98**

**General Comment:**

The study proposes a novel method of site-specific wake model calibration using SCADA data. This is a relevant area of scientific research. SCADA data is becoming more abundant and low-fidelity wake models are essential for wind farm site-assessment and control. The study first gives an overview of different wake models and the challenges in calibration. Then, the study describes the data filtering of the investigated offshore wind farm. Finally, the optimization framework and results are presented.

The presented idea for calibration is worth exploring, furthermore, the study highlights calibration problems due to the influence of neighboring farms. A problem that will become more important in the future. However, the study contains some shortcomings. Firstly, the novel calibration method itself is not the center of the paper. Instead, a lot of room is given for explanation of different wake models (some specific to the FLORIS framework) and filtering of SCADA data. The wake models have been described in many other papers. Furthermore, probably due to non-disclosure issues, the study does not describe the investigated farm, no layout is provided. This makes it much harder to visualize it for the reader.

The result section does not thoroughly analyze the results of the method. The method does not only contain the wake model calibration, but also the wind direction and wind speed adjustment which influence the results heavily. Furthermore, no comparison to a baseline model is provided.

The authors should rethink what message they want to convey with this study. *Some* possible: Is it about the optimization method? Then it should be less FLORIS-centric and give not so much room for the applied wake model. The framework should be general. Instead describe the optimization, the hyperparameter tuning and results more in detail. Is it about the uncertainty of the parameters of the Gaussian wake model? The parameters should in principle be applicable to a wide range of conditions and not change every 10 minutes. Then the authors have to describe what we learn from this volatility. Is it about the whole toolchain with filtering etc.? Then, it should be better highlighted what could be an application for the community or industry.

**Specific comments:**

Section 1.1 This section gives an overview about different wake models in FLORIS. Can the authors motivate why this section is necessary for the paper? In the end, the developed methodology should be applicable for any wake model.

line 54-55: "...there is no velocity profile..." is not correct, the wake has a top-hat velocity profile

line 55: What do the authors mean with linear wake decay? The velocity deficit does not decay linearly in the Jensen model.

line 58: "overestimates the *wake* at the edge of the wake" probably "velocity deficit" is meant here

line 60: "Gauss-legacy" is just a name of an implementation inside the FLORIS V2 package. Non-FLORIS users are not familiar what "legacy" means. On top of that, in the FLORIS V3 version (which is cited) the model is called "Gauss"

line 60. "...widely used in industry..." can you provide a reference for this statement, e.g. a survey?

line 66: "...it has become clear that traditional wake models often underestimate wake losses..." can you provide a reference for this statement?

line 68-75: Why are several wake models described? If the point is that other wake models have additional parametrizations, it would be enough to give just one example

line 76: The word "scaling" is a bit confusing and unnecessary to refer to the wake model parameters

line 79: Here it sounds like if it is a *must* to calibrate the model with CFD results and it is always done like that. However, it could be also a wind tunnel or LIDAR measurement in the first place.

line 99: What do the authors mean with "...different results were obtained."

line 109-113: It seems that in these sentences the authors actually mean the nacelle anemometer instead of wind vane? The wind vane is used to measure the wind direction. What does a bias of 5° mean for a wind speed measurement?

line 113: The wind speed also has a non-linear relation to the active power of the wind farm. What is mean with the "... calibration of the wind direction ..."

line 122 It sounds like *spatial* correction factors were applied on the wake deflection parameters which is not the case in that study.

line 136: What does "homogenoeus" mean here?

line 145-152: This paragraph just states the blockage topic. But not how it plays into the challenges of calibration through SCADA data

line 155: Can the authors motivate the use of the GCH model? As mentioned in sec. 1.1, the additional modifications are only for yaw effects, which are neglected in this study.

line 160: Can you justify not accounting for transients? Is the farm small enough or can you show that these effects are not present?

line 160-164: In the previous section (line 139ff) the authors state that stability, which is correlated to TI, is important for calibration. Can you justify neglecting it? Would it not increase the uncertainty in the calibration parameter?

line 164: "wake blockage" is not a common term. Do you mean "farm blockage"? The farm also has a blockage effect by itself.

line 171: What do the authors mean with "...essential to not skew the calibration results...".

line 171: What is prohibiting this type of analysis for binned observations?

line 190: The later part of the study suffers from the fact that the reader has no image of the farm layout. It would be helpful if the authors state the number of turbines and the type of layout (irregular/regular), average spacing.

Figure 2: The 3D depiction is a bit misleading as it could be a pie-chart that reports percentages. The figure should be clearly identifiable as a compass rose e.g. also with labels in degree

Figure 4 & Figure 5: Can the authors provide a definition of the displayed metrics? Which quantity was used for normalization?

line 245: What do the authors mean with "...above rated SCADA data..." ?

Figure 8: Normally TI is defined as std(ws)/ws so figure 6 and 8 are not very different. It is just that 6 is additionally normalized by the wind speed. Why is it important to show also figure 8?

line 265: Typo "... , The ..."

Table 1: Secondary steering as model is not necessary as no yaw is considered in this work

line 295ff.: Can the authors elaborate on the point that the sensitivity increases for specific directions? Are there higher or lower wake losses? The lack of a provided layout makes it difficult here.

Equation 8: Can the authors motivate the use of this error type? It seems that this is an error on farm level. Positive and negative turbine errors can cancel out. To improve the wake calibration, shouldn't the error be calculated on turbine level?

line 323: Can the authors state the found weighting between a and b?

line 331: The allowed range seems quite high. As the authors stated themselves, the wind speed has the largest impact on the farm power. How often did the optimization go to this extreme value? How can the estimated wind speed from the SCADA data be so far off?

Section 4.2: It would be good if the subclusters can at least be described a bit more in their configuration. Furthermore, the discussion should also include the results from a baseline model that is not optimized for comparison.

Figure 15 & 16: The figures have similar captions. It should be clear that they represent different subclusters.

line 395ff. It is not clear whether some directions were excluded from the optimization. In sec. 2.3 the authors state that directions 250-50° were excluded, yet the figure 19 (and also 15 and 16) display all directions.

Code availability: This is just the code of the open-source packages used. Not the own code developed for this study

---

## Referee Comment (RC2)

**Review of "Hyperparameter tuning framework for calibrating analytical wake models using SCADA data of an offshore wind farm applied on FLORIS"**

**General comments**

The authors present a scheme to calibrate parameters of an analytical wake model. Their method compares power predictions of the model to the power reported in the SCADA data of wind turbines and adjusts the model parameters in three steps to achieve optimal agreement. For validation, the method is applied to an offshore wind farm tuning an analytical wake model. The calibrated model parameters are then analyzed in terms of sensitivity and dependency to wind direction and wind speed.

The overall goal of the paper is to establish a new model calibration method, which is a clear and relevant agenda. However, achieving this goal is currently obstructed by issues of the specific implementation presented here and a missing comparison to other calibration methods for model parameters. In addition, there are some inaccuracies in the paper. I list my most important comments below:

(1) The model parameters are tuned with a cost function between the power predictions of the FLORIS model framework and the power reported in the SCADA data of wind turbines. However, only one part of the full FLORIS model framework, namely the velocity deficit model, has been included in the calibration. This might lead to unrealistic tuning results for the model parameters, if any of the other parts of FLORIS framework not included in the calibration is not set up optimal.
I want to illustrate the above with a specific example in the following. The methods describe the velocity deficit model, but do not provide any information how the velocity field is then related to the power. Specifically, the following points are unclear:
-   Which power coefficient and thrust coefficient curves have been used and are they realistic for the wind turbines at the test site?
-   If power curves generated from the SCADA data have been used, it would be important to know if they were applied to the model using a rotor averaged wind speed or the wind speed at the nacelle location? Were rotor blockage effects included or not?

The results of the tuning find that the wake growth rate ($k\_a$) has an optimal value that is much lower than typical literature value. However, this outcome might also be caused by a too high power coefficient set in the model framework, which the optimization then tries to correct by reducing the wake recovery.
Therefore, the found differences between the calibrated model parameters and literature values cannot be attributed to better tuning with certainty. While this does not invalidate the tuning method proposed by the authors per se, it is a problem for its validation in my opinion.

(2) Only results achieved with the here proposed calibration method are shown. It would be interesting to include results using model parameters obtained with other calibration methods and standard values from literature. This could be followed by a discussion of the differences between methods.

(3) The discussion of the results refer to many site-specific effects like farm-to-farm interaction and the influence of a nearby shoreline. However, the description of the test site is very sparse for such a discussion.

**Specific comments**

Line 43-45: This statement should be supported by a citation.

Line 45-46: It is true that measurement errors affect the characterization of the flow state, but the statement seems to be out of place at this point in the manuscript.

Line 66-75: Maybe state the history and parameters for the first model and then for the second model instead of going back and forth throughout the paragraph.

Line 90-92: The paper should motivate the proposed calibration method by pointing out benefits and differences to the already existing calibration approaches referenced here.

Line 108: It should be elaborated what stochastic uncertainty means here. Does it refer to the stochastic error of a mean wind speed that is computed from a finite ensemble of measurements of the turbulent flow? Further possible error sources that can affect it are a drift of the mean value due to diurnal cycle or changing weather conditions. In addition, the mean wind speed can have spatial variation and a single value might not be representative for a large wind farm.

Line 109-111: A wind vane measures the wind direction, but not the wind speed.

Line 130: Referring back to the comment on line 108, turbulence and noise can be adequately quantified with the variance. Other effects like a trend can lead to a non-Gaussian distribution of the measurement values. Would it be possible to extend the proposed framework with other metrics in principle?

Line 164: Does wake blockage refer to farm-to-farm interaction from neighboring wind farms or to wake effects of individual wind turbines within the wind farm?

Section 2.1 in general: The introduction of the test site does not provide sufficient information. Specifically, information should be provided on the topography of the nearby shore, the distance of the wind farm from the shore, distances to neighboring wind farms. The information provided in Figure 2 could extended with precise angles of the affected sectors and distances. Can you provide the location and a map of the wind farm and the surrounding area (if not due to NDA restrictions, that should be stated)?

Figure 3: I assume this figure only shows the filtering described in Section 2.1, but not the steps of Section 2.3 and 2.4. Is it possible to illustrate the impact of the other steps on the data? And should the values for above 25 m/s not be removed as well, because the wind turbines seem to be derating? In addition, the label on the ordinate should be normalized active power.

Line 249-250: How does the ratio of wind speed variance to wind direction variance relate to curtailment of the wind farm? To me, this not clear intuitively and should be further elaborated. I would expect that a low active power for a given wind speed is more indicative of curtailment (in the absence of status alerts or strong wake effects).

Line 257: Cite examples for the use of the model in literature here.

Section 3.1: The choice of using the term GCH might be confusing, because the Curl model outlined in Martínez-Tossas et al. (2019) is not applied here and the Gauss-legacy model is used instead. How is other literature handling the nomenclature?

Line 278-280: Can the other model components affect the results of the parameter tuning of the velocity deficit model presented here? A description of them is missing entirely.

Table 1: Abbreviation SOSFS not introduced. A column could be added to the table providing references to papers describing each of the model component.

Section 3.2: Currently, this part of the paper seems to be separated from the proposed parameter-tuning scheme. How does it contribute to the tuning of the parameters? Should users of the tuning method first conduct a sensitive analysis and remove low sensitivity parameters prior to the model calibration?

Figure 295-298: The parameters $k_a$ and $k_b$ have higher values for wind direction sectors (0, 90) and (180, 270) compared to other wind directions, while it is inverse for alpha and beta. Is there any explanation why that might be the case?

Line 325: What is the benefit of letting the algorithm choose the weighting? It would be problematic if the algorithm chooses a=0.001 and b=0.999 for example and only account for the global power production of the wind farm while the power for individual wind turbines are completely off. The results section should show what values the algorithm chose.

Line 329: What does "freeflow wind turbines" mean? Is it first turbine row of the wind farm at the upstream edge?

Line 366-371: Two questions on the interpretation here:
(1) How does the internal arrangement of the wind turbines inside the wind farm differ for the two clusters presented in Fig. 15 and Fig. 16?
(2) There does not seem to be any effect of shoreline to the south and south-east, which one might expect to also cause problems similar to a neighboring wind farm. Is it further away or is its effect on the incoming free flow less pronounced?
An overview map would be helpful to follow the interpretation of the results here. If a NDA is preventing to include it explicitly, maybe a schematic overview can be provided instead. Plotting the data as a function of the distance to the next heterogeneity upstream might another approach to avoid NDA.

Line 389-391: Isn't it the other way around? The parameter $k_a$ is not inverse to the wake recovery, because the larger $k_a$ is the larger will be the wake recovery for a given turbulence intensity.

Section 4.3 in general: As mentioned in the first general comment, I am not convinced that the tuned parameters are necessarily more realistic values, because other parts of the full FLORIS framework might interfere with the optimization. If this comment cannot be refuted directly, it might be addressed by restricting the validation to a simpler configuration (e.g. running it directly on the wind speed instead of the power (removes the power coefficient from the validation) and using first and second row turbines only (removes wake superposition from the validation)). Another approach might

be to assess the sensitivity of the tuned parameter to changes to those other parts of the FLORIS framework.

Line 403-404: There is a contradiction here. The text states that k_a has higher values for the south-east where the coast is, but in Fig. 19 the k_a values for the wind direction sector (180, 250) are lower compared to the sector (50, 180).

Line 404-405: The hypothesis of an increase of k_a with the turbulence intensity could be tested with a plot similar to Fig. 18 and Fig. 19, but with TI on the abscissa.

Captions of most figure can be improved to explain what the figure is showing without having to refer back to the text.

**Technical comments**

Line 54: Maybe rephrase instead of "no velocity profile" that the Jensen wake model assumes a constant velocity across a wake cross-section.

Line 56-57: Either "by literature (Barthelmie et al., 2009)" or "by Barthelmie et al. (2009)."

Line 123-124: Insert "are" in "effects becoming".

Line 133: Remove "significant".

Line 189-191: Sentence structure not correct.

Line 200: Remove "different".

Line 201: Mean values and variances are not measured directly, but calculated for the 10-minute intervals.

Line 202: Wind turbine should be plural here.

There are several instances where it could considered to combine separate figures into a single figure with multiple panels.

---

## Author Comment (AC1)

**Reply to Reviewers:**

We would like to thank the reviewers for their thorough review of our manuscript and for providing insightful comments and suggestions. We greatly appreciate the reviewers' constructive feedback, which has significantly enhanced the quality of our manuscript. All the comments below have been addressed accordingly and adjustments have been made to the paper.

We have carefully considered all comments and have revised the manuscript to address the concerns and suggestions raised by the reviewers. Below we provide detailed responses to each comment, outlining the changes made or our reasoning if no changes were deemed necessary. The recurring theme in the feedback concerned site confidentiality, validation, and comparison with methods in existing literature. We have addressed these comments thoroughly, and believe that our revisions will address the reviewers' concerns. We look forward to hearing your feedback.

The replies to the reviewers' comments can be found below:

**Reviewer 1**

**General comments:**
*The study proposes a novel method of site-specific wake model calibration using SCADA data. This is a relevant area of scientific research. SCADA data is becoming more abundant and low-fidelity wake models are essential for wind farm site-assessment and control. The study first gives an overview of different wake models and the challenges in calibration. Then, the study describes the data filtering of the investigated offshore wind farm. Finally, the optimization framework and results are presented.*

*The presented idea for calibration is worth exploring, furthermore, the study highlights calibration problems due to the influence of neighboring farms. A problem that will become more important in the future. However, the study contains some shortcomings. Firstly, the novel calibration method itself is not the center of the paper. Instead, a lot of room is given for explanation of different wake models(some specific to the FLORIS framework) and filtering of SCADA data. The wake models have been described in many other papers. Furthermore, probably due to non-disclosure issues, the study does not describe the investigated farm, no layout is provided. This makes it much harder to visualize it for the reader.*

*The result section does not thoroughly analyze the results of the method. The method does not only contain the wake model calibration, but also the wind direction and wind speed adjustment which influence the results heavily. Furthermore, no comparison to a baseline model is provided.*

*The authors should rethink what message they want to convey with this study. Some possible: Is it about the optimization method? Then it should be less FLORIS-centric and give not so much room for the applied model. The framework should be general. Instead describe the optimization, the hyperparameter tuning and results more in detail. Is it about the uncertainty of the parameters of the Gaussian wake model? The parameters should in principle be applicable to a wide range of conditions and not change every 10 minutes. Then the authors have to describe what we learn from this volatility. Is it about the whole toolchain with filtering etc.? Then, it should be better highlighted what could be an application for the community or industry.*

**Responses to the general comments by the authors:**

We acknowledge the reviewer's observation regarding the paper's focus. The primary intent of our research is to emphasize the holistic approach of applying hyperparameter tuning algorithms to calibrate parameters of analytical wake models using SCADA data. We regard SCADA data cleaning as an essential step before the tuning process, which is why it has been carefully described in our methodology.

We have reduced the extensive overview of the wake models, especially regarding the description of the tuning parameters available within the FLORIS framework, making the work less FLORIS-centric.

We understand the importance of visualizing the wind farm layout and surroundings for the reader. We were initially limited in the details we could provide. However, we have incorporated more general information about the wind farm's surroundings and topology.

The feedback prompted us to re-evaluate the core message of the paper. As mentioned, our primary emphasis is on the holistic context of hyperparameter tuning using SCADA data. We hope our revised manuscript better aligns with this focus.

**Specific comments:**

1. **Reviewer:** *Section 1.1: This section gives an overview about different wake models in FLORIS. Can the authors motivate why this section is necessary for the paper? In the end, the developed methodology should be applicable for any wake model.*
   **Author:** Thank you for raising the concern about the relevance of Section 1.1. The intention with this section was to introduce the reader to the evolution and structure of analytical wake models throughout the years, making it more accessible to readers less familiar with these models. We agree with the reviewer that the last Paragraph is less relevant to the paper's core focus and is therefore removed. Furthermore, adjustments to the title are made to further emphasize the purpose of this section.

2. **Reviewer:** *Line 54-55:"…there is no velocity profile…" is not correct, the wake has a top-hat velocity profile*
   **Author:** Correct, this has been adjusted.

3. **Reviewer:** *What do the authors mean with linear wake decay? The velocity deficit does not decay linearly in the Jensen model.*
   **Author:** Correct, the velocity decays proportionally to the inverse of the distance from the turbine, while the wake width expands linearly. This has been adjusted.

4. **Reviewer**: *"Overestimates the wake at the edge of the wake" probably "velocity deficit" is meant here*
   **Author**: Corrected.

5. **Reviewer**: *Line 60: "Gauss-Legacy" is just a name of an implementation inside the FLORIS V2 package. Non-FLORIS users are not familiar what "legacy" means. On top of that, in the FLORIS V3 version (which is cited) the model is called "Gauss".*
**Author**: We are aware of this, and this has been adjusted accordingly.

6. **Reviewer**: *Line 66: "… it has become clear that traditional wake models often underestimate wake losses…" can you provide a reference for this statement?*
**Author**: A reference has been provided now.

7. **Reviewer**: *Line 68-75: Why are several wake models described? If the point is that other wake models have additional parameterizations, it would be enough to give just one example*
**Author**: The inclusion of multiple wake models was not intended to highlight additional parameters specifically, but rather to provide an overview of the wake models available in the literature and explain why they have been developed.
Since the analysis has been conducted on a large-scale wind farm exceeding 300MW capacity, it is important to determine if the reports, which mention that the Jensen and Gauss wake models underestimate the cluster-wake losses, agree with the findings of this work (e.g. if its due to the nature of the analytical Equations or due to wrong calibration). When we as authors, mention this statement, we feel it leaves a gap to not mention developments that have been done to mitigate these cluster-wake losses, and therefore to create a holistic picture this has been mentioned.
We acknowledge that lines 71-75 do not add much value to the section, and therefore we have removed this part of the text.

8. **Reviewer**: *Line 76: The word "scaling" is a bit confusing and unnecessary to refer to the wake model parameters*
**Author**: Corrected.

9. **Reviewer**: *Line 79: Here it sounds like if it is a must to calibrate the model with CFD results and it is always done like that. However, it could be also a wind turnnel or LIDAR measurements in the first place.*
**Author**: That is indeed not the case. This has been corrected.

10. **Reviewer**: *Line 99: What do the authors mean with "…different results were obtained."*
**Author**: Different parameter sets were obtained from calibration

11. **Reviewer**: *Line 109-113: It seems that in these sentences the authors actually mean the nacelle anemometer instead of the wind vane? Wind vane is used to measure the wind direction. What does a bias of 5 degrees mean for a wind speed measurement.*
**Author**: Corrected.

12. **Reviewer**: *Line 113: The wind speed also has a non-linear relation to the active power of the wind farm. What is meant with the "…calibration of the wind direction…"*
**Author**: That is correct. What we meant is that when determining active power based on wind speed below rated wind speed using a cost function, there are no local minima.

This is not the case for the wind direction, where local minima might be present. This can be observed when examining energy ratios for example. By 'calibration' we mean 'determining'. We've made adjustments for better understanding.

13. **Reviewer**: *Line 122: It sounds like spatial correction factors were applied on the wake deflection parameters, which is not the case in that study.*
    **Author**: Corrected.

14. **Reviewer**: *Line 136: What does "homogeneous" mean here?*
    **Author**: Horizontally homogeneous inflow.

15. **Reviewer**: *Line 145-152: This paragraph just states the blockage topic. But not how it plays into the challenges of calibration through SCADA data*
    **Author**: Blockage, when overlooked, can introduce additional complexities in SCADA data interpretation, especially in large wind farms. The blockage effect could adjust the observed wind speed and wind direction. For the specific wind farm in question, which is part of a large cluster, modeling the blockage would present significant challenges. Fortunately, we did not observe any indications of spatially varying wind directions attributable to blockage for this farm. However, considering the larger picture including neighbouring wind farms, blockage cannot be ignored. Therefore, we felt it was necessary to address this in our study.

16. **Reviewer**: *Line 155: Can the authors motivate the use of the GCh model? As mentioned in sec. 1.1, the additional modifications are only the yaw effects, which are neglected in this study.*
    **Author**: Previous studies regarding tuning are generally performed on Gaussian models, where the latest tuning paper by Beek et al. (2021) performed the calibration on the Gauss-Curl Hybrid model. The GCH modifications are indeed mainly due to secondary steering, but also include wake asymmetry due to wake rotation, as mentioned in King et al. (2019). Therefore, we considered the GCH model, instead of the Gaussian model without secondary-steering and wake asymmetry.

17. **Reviewer**: *Line 160: Can you justify not accounting for transients? Is the farm small enough or can you show that these effects are not present?*
    **Author**: If you refer to 'no alternations are made to account for temporal variability', then we justify this by accurate filtering of the SCADA data. By doing so we have effectively removed data that shows large temporal variations, such as those arising from changing weather conditions.

18. **Reviewer**: *Line 160-164: In the previous section (line 139ff) the authors state that stability, which is correlated to TI, is important for calibration. Can you justify neglecting it? Would it not increase the uncertainty in the calibration parameter?*
    **Author**: Not considering TI and/or stability does indeed increase uncertainty. While these factors are indeed important, a detailed analysis on them is planned for separate, focused research, as it extends beyond the scope of this paper.

19. **Reviewer**: *Line 164: "wake blockage" is not a common term. Do you mean "farm blockage"? The farm also has a blockage by itself.*
    **Author**: Corrected.

20. **Reviewer**: *Line 171: What do authors mean with :…essential to not skew the calibration results…".*
    **Author**: Accurately determining freeflow wind speed and wind direction in a wind farm is crucial as errors can skew calibration results. Specifically, overestimating the freeflow wind speed can exaggerate wake effects, while underestimating the freeflow wind speed can do the opposite. Most literature uses a cost-function matching SCADA and model active power, which is sensitive to such discrepancies. Our solution incorporates the wind speed into the calibration process for time series data, ensuring minimal skewing of results, and avoiding under-or-overestimated due to wind speed mismatches. The same logic applies to wind direction calibration.

21. **Reviewer**: *Line 171: What is prohibiting this type of analysis for binned observations?*
    **Author**: Binned analysis assumes balance: It is valid when the magnitude and frequency of overestimation are in balance with the frequency of underestimations. Otherwise, results can be skewed. Additionally, the volume of usable data becomes limited in binned observations, since even the downtime of a single turbine can introduce significant skewing.

22. **Reviewer**: *Line 190: The later part of the study suffers from the fact that the reader has no image of the farm layout. It would be helpful if the authors state the number of turbines and the type of layout (irregular/regular), average spacing.*
    **Author**:  We have addressed this by providing a detailed overview of the case study wind farm surroundings in Figure 2. Furthermore, we have specified the type of layout in the relevant section.

23. **Reviewer**: *Figure 2: The 3D depiction is a bit misleading as it could be a pie-chart that reports percentages. The figure should be clearly identifiable as a compass rose e.g. also with labels in degree*
    **Author**: We have revised Figure 2 to give a better representation of the case study wind farm surroundings.

24. **Reviewer**: *Figure 4 & 5: Can the authors provide a definition of the displayed metrics? Which quantity was used for normalization?*
    **Author**: We have now added the Equations to the Figures.

25. **Reviewer**: *Line 245: What do authors mean with "…above rated SCADA data…"?*
    **Author**: "…the SCADA data associated with the above-rated wind speed region…". We did adjust this in the paper.

26. **Reviewer**: *Figure 8: Normally TI is defined as std(ws)/ws so figure 6 and 8 are not very different. It is just that 6 is additionally normalized by wind seed. Why is it important to show also figure 8?*
    **Author**: Figure 8 helps in the general understanding of Section 2.4. Here a filter is applied based on the fraction of the wind speed and wind direction variance. Figure 7 illustrates that the variance in wind direction tends to increase with increasing wind speeds, while the variance in wind speed tends to rise. We have provided some additional explanation on why we use this type of filtering within the paper.

27. **Reviewer**: *Line 265: Typo "…, The…"*
    **Author**: Corrected.

28. **Reviewer**: *Table 1: Secondary steering as model is not necessary as no yaw is considered in this work*
    **Author**: Correct, it has been added since it was part of the settings, but it can indeed be removed.

29. **Reviewer**: *Line 295ff.: Can the authors elaborate on the point that the sensitivity increases for specific directions? Are there higher or lower wake losses? The lack of provided layout makes it difficult here.*
    **Author**: See Figure 2 and 15. The wind farm is rectangularly shaped, with a spacing denser from SE<-> NW compared to SW <-> NE. While we cannot disclose the full topology, based on the layout, we hypothesize that parameters alpha and beta are particularly significant in areas where turbines have limited spacing, while parameters ka and kb gain importance in areas with greater spacing and a larger number of clustered turbines.

30. **Reviewer**: *Equation 8: Can the authors motivate the use of this error type? It seems that this is an error on farm level. Positive and negative turbine errors can cancel out. To improve the wake calibration, shouldn't the error be calculated on turbine level?*
    **Author**: Equation 8 represents the farm-level error. The turbine-level error is determined using Equation 7, with weights a and b emphasizing the importance of Equations 7 and 8, respectively.

31. **Reviewer**: *Line 323: Can the authors state the found weighting between a and b?*
    **Author**: Certainly, this section has been updated. To ensure that a remains independent of the turbine count, $1/N\_T$ has been excluded from function g(), as shown in eq. 8. This adjustment ensures that consistent scalar values for a and b can be chosen irrespective of the number of turbines.

32. **Reviewer**: *The allowed range seems quite high. As the authors stated themselves, the wind speed has the largest impact on the farm power. How often did the optimization go to this extreme value? How can the estimated wind speed from the SCADA data be so far off?*
    **Author**: You are correct to observe that the range for the wind speed is large. Our analysis of the SCADA data highlighted several instances where the wind speed from the

nacelle anemometer did not match the active power from SCADA data. Given the nacelle anemometer's location behind the rotor, this tends to introduce uncertainties in the measured speed. We believe that using the active power from the SCADA data provides a more reliable estimate of the free-flow wind speed.

While the allowed range might appear high, the absence of local minima in the wind speed-only optimization ensures this is not problematic. Furthermore, the wind speed-only optimization time is negligible. However, as pointed out by the reviewer, these extreme values are seldom reached.

33. **Reviewer**: *Section 4.2: It would be good if the subclusters can at least be described a bit more in their configuration. Furthermore, the discussion should also include the results from a baseline model that is not optimized for comparison.*
    **Author**: A Figure has been added that shows the coordinates of the wind turbines within the farm.

34. **Reviewer**: *Figure 15 & 16: The figures have similar captions. It should be clear that they represent different subclusters.*
    **Author**: Noted and updated.

35. **Reviewer**: *Line 395ff: It is not clear whether some directions were excluded from the optimization. In sec 2.3 the authors state that direction 250-50 deg were excluded, yet the figure 19 (and also 15, 16) display all directions.*
    **Author**: Our time-series based optimization allows ad-hoc wind direction filtering. For Figures 18 and 19, for example, ad-hoc filtering has been performed to showcase the freeflow conditions. On the other hand, Figures 20-23 show all wind directions, highlighting the effect of external wakes on the optimization results. In Figures 15 and 16 our objective was to highlight discrepancies in energy ratio mismatches from neighbouring wind farms, especially when turbines are in close proximity of these wind farms. Therefore, for wind directions from neighbouring wind farms are considered for Figures 15,16, 20-23.

36. **Reviewer**: *This is just the code of the open-source packages used. Not the own code developed for this study.*
    **Author**: We have updated this Section, so those interested can reach out to the corresponding author if they are interested in the code implementation.

**Reviewer 2**

**General comments:**

*The authors present a scheme to calibrate parameters of an analytical wake model. Their method*
*compares power predictions of the model to the power reported in the SCADA data of wind turbines*
*and adjusts the model parameters in three steps to achieve optimal agreement. For validation, the*
*method is applied to an offshore wind farm tuning an analytical wake model. The calibrated model*
*parameters are then analyzed in terms of sensitivity and dependency to wind direction and wind speed.*
*The overall goal of the paper is to establish a new model calibration method, which is a clear and*
*relevant agenda. However, achieving this goal is currently obstructed by issues of the specific*
*implementation presented here and a missing comparison to other calibration methods for model*
*parameters. In addition, there are some inaccuracies in the paper. I list my most important comments below:*

*(1) The model parameters are tuned with a cost function between the power predictions of the*
*FLORIS model framework and the power reported in the SCADA data of wind turbines. However,*
*only one part of the full FLORIS model framework, namely the velocity deficit model, has been*
*included in the calibration. This might lead to unrealistic tuning results for the model*
*parameters, if any of the other parts of FLORIS framework not included in the calibration is not*
*set up optimal. I want to illustrate the above with a specific example in the following. The*
*methods describe the velocity deficit model, but do not provide any information how the velocity*
*field is then related to the power. Specifically, the following points are unclear:*

- *Which power coefficient and thrust coefficient curves have been used and are they*
  *realistic for the wind turbines at the test site?*
- *If power curves generated from the SCADA data have been used, it would be*
  *important to know if they were applied to the model using a rotor averaged wind*
  *speed or the wind speed at the nacelle location? Were rotor blockage effects*
  *included or not?*

*The results of the tuning find that the wake growth rate ($k\_a$) has an optimal value that is much*
*lower than typical literature value. However, this outcome might also be caused by a too high power*
*coefficient set in the model framework, which the optimization then tries to correct by reducing the*
*wake recovery.*
*Therefore, the found differences between the calibrated model parameters and literature values*
*cannot be attributed to better tuning with certainty. While this does not invalidate the tuning*
*method proposed by the authors per se, it is a problem for its validation in my opinion.*
*(2) Only results achieved with the here proposed calibration method are shown. It would be*
*interesting to include results using model parameters obtained with other calibration methods*
*and standard values from literature. This could be followed by a discussion of the differences*
*between methods.*
*(3) The discussion of the results refer to many site-specific effects like farm-to-farm interaction and*

*the influence of a nearby shoreline. However, the description of the test site is very sparse for such a*
*discussion.*

**Responses to the general comments by the authors:**

Thank you for your detailed review and constructive feedback. We acknowledge the points raised and aim to address them comprehensively below.

1) **Model parameters within the model framework**: We appreciate the concerns regarding the specific components of the framework used in the calibration. Our decision to focus on the velocity deficit model and to exclude others, like the wake-added turbulence model, comes from the interdependencies these models show. Including both submodels could lead to a multimodal solution space, where certain parameters in one submodel influencing adjustments in another. There are solutions to mitigate this behaviour, like singular value decomposition, however, for the scope of this work only the wake velocity deficit model is considered.

   As for uncertainties related to the power curve, we believe active power measurements are more reliable than wind speed measurements, especially given that anemometer measurements are taken behind the wind turbine, which introduces uncertainties. The study employs a steady-state framework, which favors measurements with minimal temporal variations. Additionally, both thrust and power curves used are directly sourced from the operators, ensuring authenticity. We have also validated the power curve for each turbine within the studied wind farm.

   In terms of model parameters, the results were derived using a robust combination of filtering, optimization, and multiple validation stages, including energy ratios, sensitivity studies, and a Pearson correlation matrix. We have full confidence in the validity of the acquired results due to this comprehensive approach. Nonetheless, we are aware of the potential influences of other model components on the parameter tuning, such as the turbulence intensity, wake combination models, and turbulence parameters. Again, given the scope and length of this manuscript, a more in-depth analysis of these parameters is planned in the future.

2) **Comparison with other calibration methods**: We understand the concerns regarding the specific implementation and the absence of a comparison with other calibration methods. Our primary focus is on the holistic context of calibration analytical wake models using SCADA data. Given the scope and the current length of the manuscript, we plan to compare the developed method in future work.

3) **Discussion of site-specific effects**: We understand the need for a more detailed description of the case study wind farm. To address this, we have updated Figure 2 and added Figure 15. These changes provide readers with a clearer picture of the wind farm, thereby laying a stronger foundation for the discussion section of the paper.

**Specific comments:**

1. **Reviewer**: *Line 43-45: This statement should be supported by a citation.*
   **Author**:  A citation has now been included.

2. **Reviewer**: *Line 45-46: It is true that measurement errors affect the characterization of the flow state, but the statement seems to be out of place at this point in the manuscript.*
   **Author**: This sentence has been removed from this Section and added to the Section 1.3.

3. **Reviewer**: *Line 66-75: Maybe state the history and parameters for the first model and then for the second model instead of going back and forth throughout the manuscript.*
   **Author**: Reviewer 1 also mentioned that this piece of text seemed redundant. Therefore, it has been partly removed.

4. **Reviewer**: *Line 90-92: The paper should motivate the proposed calibration method by pointing out benefits and differences to the already existing calibration approaches references here.*
   **Author**: We added benefits and differences at the end of Section 1.2.

5. **Reviewer**: *Line 108: It should be elaborated what stochastic uncertainty means here. Does it refer to the stochastic error of a mean wind speed that is computed from a finite ensemble of measurements of the turbulent flow? Further possible error sources that can affect it are a drift of the mean value due to diurnal cycle or changing weather conditions. In addition, the mean wind speed can have spatial variation and a single value might not be representative for a large wind farm.*
   **Author**: By stochastic uncertainty, we primarily refer to variability and random fluctuations in wind characteristics at any frequency scale. In our case, this includes turbulence, changes in weather patterns, and the diurnal and annual cycles. A drift of the mean value due to diurnal cycles and changing weather patterns can indeed be a source of error. We've added some additional text to define stochastic uncertainty here.

6. **Reviewer**: *Line 109-111: A wind vane measures the wind direction, but not the wind speed.*
   **Author**: Corrected.

7. **Reviewer**: *Line 130: Referring back to the comment on line 108, turbulence and noise can be adequately quantified with the variance. Other effects like a trend can lead to a non-Gaussian distribution of the measurement values. Would it be possible to extend the proposed framework with other metrics in principle?*
   **Author**: Incorporating trends, such as shifting weather patterns or changes due to diurnal cycles as you pointed out, is feasible with an extended framework. However, given the scope and length of this work, and the potential for ad-hoc implementation, this will be considered for future work.

8.  **Reviewer**: *Line 164: Does wake blockage refer to farm-to-farm interaction from neighbouring wind farms or to wake effects of individual wind turbines within the wind farm?*
    **Author**: Wake blockage refers to the global blockage effect of a wind farm, causing a deflection of the wind speed upwards and sideways. We adjusted 'wake blockage' to 'wind farm blockage' for better clarity.

9.  **Reviewer**: *Section 2.1 in general: The introduction of the test site does not provide sufficient information. Specifically, information should be provided on the topography of the nearby shore, the distance of the wind farm from the shore, distances to neighbouring wind farms. The information provided in Figure 2 could extend with precise angles of the affected sectors and distances. Can you provide the location and a map of the wind farm and the surrounding area (if not due to NDA restrictions, that should be stated)?*
    **Author**: Figure 2 has been replaced by a more detailed figure, showcasing the locations of neighbouring wind farms, and the distance from the shoreline.

10. **Reviewer**: *Figure 3: I assume the figure only shows the filtering described in Section 2.1, but not the steps of Section 2.3 and 2.4. Is it possible to illustrate the impact of the other steps on the data? And should the values for above 25m/s not be removed as well, because the wind turbines seem to be derating? In addition, the label on the ordinate should be normalized active power.*
    **Author**: Figure 3 primarily illustrates the filtering process from Section 2.1. Displaying filters from Section 2.3 and 2.4 would not significantly alter the figure's appearance. The power curve and thrust curve can both consider derated conditions. However, since data beyond a certain wind speed is not considered for calibration, no significant change would occur. We have adjusted the x-label accordingly.

11. **Reviewer**: *Line 249-250: How does the ratio of wind speed variance to wind direction variance relate to curtailment of the wind farm? To me, this not clear intuitively and should be further elaborated. I would expect that a low active power for a given wind speed is more indicative of curtailment (in the absence of status alerts or strong wake effects).*
    **Author**: We have addressed most curtailment issues through Section 2.1. However, when slight discrepancies occur between active power and wind speed above rated performance, the stage 1 optimization can lead to a lower freeflow wind speed than the actual freeflow wind speed. Another indicator of large wind speeds is the variance ratio between wind direction and wind speed. Filtering based on this ratio effectively removes timestamps that otherwise result in a mismatch in freeflow wind speed. Through an iterative test, it has been observed that a ratio of 1/40 works best to minimize the number of mismatches.

12. **Reviewer**: *Line 257: Cite examples for the use of the model in Literature here.*
    **Author**: This has been addressed.

13. **Reviewer**: *Section 3.1: The choice of using the term GCH might be confusing, because the Curl model outlined in Martinez-Tossas et al. (2019) is not applied here and the Gauss-legacy model is used instead. How is other literature handling the nomenclature.*
    **Author**: While the curl-model from Martinez-Tossas et al. (2019) is not directly applied, its introduction of a slight asymmetric wake due to counter-rotating vortices is incorporated. We have considered using the name 'gauss-legacy' velocity deficit model in combination with the effect of counter-rotating vortices, but we feel that referring to it as GCH offers a more straightforward understanding of the model type. In light of feedback from Reviewer 1, we have also made modifications to the term 'gauss-legacy' in Table 1, since it referred to FLORIS V2 instead of FLORIS V3.

14. **Reviewer**: *Line 278-280: Can the other model components affect the results of the parameter tuning of the velocity deficit model presented here? A description of them is missing entirely.*
    **Author**: Other model components can influence the parameter tuning, such as TI and the type of wake combination model. Our framework doesn't optimize the wake-added turbulence model parameters, due to the possible occurrence of multiple optimal solutions. However, we use the standard values in the literature, as defined in FLORIS V3 and Crespo Hernandez (1996). Detailed analysis of these parameters is reserved for future work.

15. **Reviewer**: *Table 1: Abbreviation SOSFS not introduced. A column could be added to the table providing references to papers describing each of the model components.*
    **Author**: We have updated the table to now include the necessary citations.

16. **Reviewer**: *Section 3.2: Currently, this part of the paper seems to be separated from the proposed parameter-tuning scheme. How does it contribute to the tuning parameters? Should users of the tuning method first conduct a sensitive analysis and remove low sensitivity parameters prior to the model calibration?*
    **Author**: A sensitivity study like this is indeed crucial. For the considered Gaussian model, all parameters retain some sensitivity. However, some models can include parameters with little to no sensitivity, meaning those can potentially be removed from the optimization process. Furthermore, knowing the sensitivity helps assess the validity of results, especially when considering potential correlations.

17. **Reviewer**: *Figure 295-298: The parameters $k\_a$ and $k\_b$ have higher values for wind direction sectors (0,90) and (180,270) compared to other wind directions, while it is the inverse for alpha and beta. Is there any explanation why that might be the case?*
    **Author**: The layout of the wind farm, now shown in Figures 2 and 15, should clarify this; Given the wind farm's rectangular shape, it is hypothesized that ka and kb display more sensitivity in cluster wakes with larger spacing, while alpha and beta gain significance when turbines are situated closer together additionally with less cluster wakes.

18. **Reviewer**: *Line 325: What is the benefit of letting the algorithm choose the weighting? It would be problematic if the algorithm chooses a = 0.001 and b = 0.999 for example and only account for the global power production of the wind farm while the power for*

*individual wind turbines are completely off. The results section should show what values the algorithm chose.*
**Author**: This has now been addressed in the manuscript and explained in bulletpoint 31 for reviewer 1.

19. **Reviewer**: *Line 329: What does "freeflow wind turbines" mean? Is it first turbine row of the wind farm at the upstream edge?*
    **Author**: Correct, we adjusted this in the paper.

20. **Reviewer**: *Line 366-371: Two questions on the interpretation here:*
    1) *How does the internal arrangement of the wind turbines inside the wind farm differ for the two clusters presented in Fig. 15 and Fig 16?*
    2) *There does not seem to be any effect of shoreline to the south and south-east, which one might expect to also cause problems similar to a neighbouring wind farm. Is it further away or is its effect on the incoming flow less pronounced?*

    *An overview map would be helpful to follow the interpretation of the results here. If a NDA is preventing to include it explicitly, maybe a schematic overview can be provided instead. Plotting the data as a function of distance to the next heterogeneity upstream might be another approach to avoid NDA.*

    **Author**: With the addition of Figures 2 and 15, the layout should now be clearer. Wind from neighbouring wind turbines causes heterogeneous inflow within the wind farm, which the Framework with homogeneous inflow cannot account for. The shoreline, being far away and regular, mainly influences the wind speed, resulting in a possible speed-up scenario, which is compensated for by the model.

21. **Reviewer**: *Line 389-391: Isn't it the other way around? The parameter k_a is not the inverse of wake recovery, because a larger k_a is the larger will be the wake recovery for a given turbulence intensity.*
    **Author**: You are right, this oversight has been corrected.

22. **Reviewer**: *Section 4.3 in general: As mentioned in the first general comment, I am not convinced that the tuned parameters are necessarily more realistic values, because other part of the full FLORIS framework might interfere with the optimization. If this comment cannot be refuted directly, it might be addressed by restricting the validation to a simpler configuration (e.g. running it directly on the wind speed instead of the power (removes the power coefficient from the validation) and using first and second row turbines only (removes wake superposition from the validation)). Another approach might be to assess the sensitivity of the tuned parameters to changes to those parts of the FLORIS framework.*
    **Author**: We appreciate your feedback and understand your concerns. Some of these points have already been addressed in our initial response, but let me elaborate it further for clarity:

**Power curve vs. wind speed**: Active power measurements, in our experience and based on data at hand, are notably more reliable than wind speed measurements since the anemometer sensor is placed behind the wind turbine blades, which therefore skews the measurement. Furthermore, given the limited inertia of an anemometer, can result in large time-based fluctuations. When a steady-state Framework is used, stable measurements such as active power, tend to provide more consistent results.
**Validity of power and thrust curves**: The curves used within this work are the official curves from the turbine operators and are validated for each turbine within the wind farm.
**Simple configuration**: While restricting our optimization to the first and second row of turbines might simplify matters, it will significantly narrow the scope of our insights. Our objective is to gain a holistic, real-world applicable understanding, and we believe this is only achievable when the methodology is applicable to all wind turbines within a wind farm.
**Sensitivity:** Addressing your feedback on the sensitivity of the tuned parameters, we have plans to further analyze the effect of additional parameters within the FLORIS framework on the obtained results. This will primarily focus on aspects susch as the combination model and the wake-added turbulence model. We furthermore aim to compare different frameworks, using the same optimization method presented in this paper. For this work, we believe our validation, grounded in a sensitivity study of the parameters, a robust filtering stage and additional checks, provides a solid foundation for our conclusions.

23. **Reviewer**: *Line 403-404: There is a contradiction here. The text states that k_a has higher values for south-east where the coast is, but in Fig. 19 the k_a values for the wind direction sector (180-250) are lower compared to sector (50-180).*
    **Author**: South-east is the coastal area (e.g. 120-150), where we see a clear peak for ka in Figures 19 and 20. Sector 180-250 is predominantly wind from sea and therefore results in lower values of ka.

24. **Reviewer**: *Line 404-405: The hypothesis of an increase in k_a with the turbulence intensity could be tested with a plot similar to Fig.18 and Fig.19 but with TI on the abscissa.*
    **Author**: This is a valuable suggestion. We plan to analyze this in future work in combination with stability indicators, and annual and diurnal cycles.

25. **Reviewer**: *Captions of most figure can be improved to explain what the figure is showing without having to refer back to the text.*
    **Author**: We have updated and extended most of the figure captions for better clarity.

**Technical comments:**

26. **Reviewer**: *Line 54: maybe rephrase instead of "no velocity profile" that the Jensen wake model assumes a constant velocity across a wake cross-section.*
    **Author**: Adjusted.

27. **Reviewer**: *Line 56-57: Either 'by literature (Barthelmie et al., 2009)' or 'by Barthelmie et al. (2009)'*
    **Author**: Corrected.

28. **Reviewer**: *Line 123-124: Insert 'are' in 'effects becoming'*
    **Author**: Corrected.

29. **Reviewer**: *Line 133: Remove 'significant'*
    **Author**: Removed.

30. **Reviewer***: Line 189-191: Sentence structure not correct*.
    **Author**: Adjusted.

31. **Reviewer**: *Line 200: Remove 'different'*
    **Author**: Removed.

32. **Reviewer***: Line 201: Mean values and variances are not measured directly, but calculated for 10-minute intervals*
    **Author**: Corrected.

33. **Reviewer**: *Line 202: Wind turbine should be plural here*
    **Author**: Corrected.

34. **Reviewer***: There are several instances where it could considered to combine separate figures into a single figure with multiple panels*
    **Author**:

---

## Referee Report (RR1)

**Review comments on revised manuscript for wes-2023-98**

The authors have adressed the reviewers comments (except see last comment). The paper is recommended for publication after minor revisions:

**Reviewer (first review):** Line 145-152: This paragraph just states the blockage topic. But not how it plays into the challenges of calibration through SCADA data
**Author:** Blockage, when overlooked, can introduce additional complexities in SCADA data interpretation, especially in large wind farms. The blockage effect could adjust the observed wind speed and wind direction. For the specific wind farm in question, which is part of a large cluster, modeling the blockage would present significant challenges. Fortunately, we did not observe any indications of spatially varying wind directions atributable to blockage for this farm. However, considering the larger picture including neighbouring wind farms, blockage cannot be ignored. Therefore, we felt it was necessary to address this in our study.
**Reviewer:** Again, blockage could also occur from a single farm. Therefore the argument "we did not observe any indications of spatially varying wind directions atributable to blockage for this farm" should also be taken into the text.

**Reviewer (first review):** Line 171: What is prohibiting this type of analysis for binned observations?
**Author:** Binned analysis assumes balance: It is valid when the magnitude and frequency of overestimation are in balance with the frequency of underestimations. Otherwise, results can be skewed. Additionally, the volume of usable data becomes limited in binned observations, since even the downtime of a single turbine can introduce significant skewing.
**Reviewer:** This reasoning should also be reflected in the paper text.

**Reviewer (first review):** Figure 4 & 5: Can the authors provide a definition of the displayed metrics?
Which quantity was used for normalization?
**Author:** We have now added the Equations to the Figures
**Reviewer:** Technical suggestion: Because of the fraction you could put the definitions not in the figure caption but in the paper text where there is more space to introduce them.

**Reviewer (first review):** Section 4.2: It would be good if the subclusters can at least be described a bit more in their configuration. Furthermore, the discussion should also include the results from a baseline model that is not optimized for comparison.

**Author:** A Figure has been added that shows the coordinates of the wind turbines within the farm.

**Reviewer:** An answer to the second part of the comment was given to the other reviewer. The authors have decided against comparing their results to an uncalibrated/different model. In my opinion this is of critical importance in the future to benchmark the proposed method

---

## Author Response (AR2)

**Reply to Reviewers:**

We would like to thank the reviewers for their comprehensive review and insightful comments, as well as to the associate editor for highlighting important concerns regarding our manuscript. In response, we have addressed each point raised and we believe the paper is now fundamentally better than before. Below, we provide the detailed responses to each comment, and we hope that our revisions addressed all the concerns which the reviewers and the associate editor had.

**Associate editor:**

***Associate editor:*** *Thank you for your revisions to the previous version of the manuscript. As the reviewers indicate, many of the initial concerns have been addressed. However, the following concerns should still be addressed.*

1. **From Reviewer 1***: regarding the possibility of blockage – reasoning on excluding the blockage from consideration should be incorporated into the text, perhaps using the response to reviewers, 'we did not observe any indications of spatially varying wind directions attributable to blockage for this farm.'*
   **Author response:** We have now incorporated a portion of text mentioning the reason why a blockage model is not incorporated within the case study wind farm.

2. **From Reviewer 1***: regarding binning, please also incorporate the reasoning for avoiding binning into the manuscript text.*
   **Author response:** We have now incorporated the reasoning for avoiding binning into the manuscript text.

3. **From Reviewer 1***: a more thorough justification is required of the choice to not compare results against an uncalibrated model.*
   **Author response:** We have decided to incorporate a comparison of results against the uncalibrated model. This comparison can be observed in Figure 15, with some additional text explaining the error metric. The error metric is based on the metric applied in Nygaard N. G. et al., (2022). We updated Figure 9 and Figure 10 to comply with the same metric, as described using Equations 3 and 4

4. **From Reviewer 2***: a more thorough discussion is required of contributing factors (and resulting limitations of this study) that cause differences between the relationship of wake growth rate and turbulence intensity seen here and the rest of the literature. (See Reviewer 2's extensive first concern.)*
   **Author response:** We have now incorporated this into our paper, discussing the factors contributing to the observed differences between the wake growth rate compared to existing literature. Here we mainly focus on the turbulence intensity and the scale difference of our cost-function metric when compared to the rest of the literature. Furthermore, we acknowledge the limitation of assuming a constant turbulence intensity in our conclusion section, where we also highlight the potential impact of site-specific characteristics on the calibration outcomes. Consequently, we explicitly advice against the direct application of our calibrated tuning parameters to different sites without further recalibration.

5. **From Reviewer 2**: *A thorough discussion of the filtering criteria used to justify the approach.*
   **Author response:** We identified that this additional filtering criteria was necessary due to either above-rated curtailment, reaching values close to rated power production, or minor underpredictions of power prediction at rated capacity. It became clear that a one-size-fits-all approach might not be suitable for every wind farm, therefore we tried to explore alternative methods. Consequently, we have refined the power-curve filtering process to more precise exclude data points related to curtailment near the rated power. We use the method proposed by Doekemeijer, Simley and Fleming (2022) with stricter margins. This can be observed in Figure 3.

Additionally, we have considered the remaining reviewers' comments as follows:

**Reviewer 1:**

1. **Reviewer (first review)**: *Line 145 – 152: This paragraph just states the blockage topic. But not how it plays into the challenges of calibration through SCADA data.*
   **Author**: Blockage, when overlooked, can introduce additional complexities in SCADA data interpretation, especially in large wind farms. The blockage effect could adjust the observed wind speed and wind direction. For the specific wind farm in question, which is part of a large cluster, modeling the blockage would present significant challenges. Fortunately, we did not observe any indications of spatially varying wind directions attributable to blockage for this farm. However, considering the larger picture including neighbouring wind farms, blockage cannot be ignored. Therefore, we felt it was necessary to address this in our study.
   **Reviewer**: *Again, blockage could also occur from a single wind farm. Therefore, the argument "we did not observe any indications of spatially varying wind directions attributable to blockage for this farm" should also be taken into the text.*
   **Author response**: We have now described this within the text (see author response 1 to the associate editor).

2. **Reviewer (first review):** *Line 171: What is prohibiting this type of analysis for binned observations?*
   **Author**: Binned analysis assumes balance: It is valid when the magnitude and frequency of overestimation are in balance with the frequency of underestimations. Otherwise, results can be skewed. Additionally, the volume of usable data becomes limited in binned observations, since even the downtime of a single turbine can introduce significant skewing.
   **Reviewer**: *This reasoning should also be reflected in the paper text.*
   **Author response**: This has now been added to the paper text (see author response 2 to the associate editor).

3. **Reviewer (first review):** *Figure 4 & 5: Can the authors provide a definition of the displayed metrics? Which quantity was used for normalization?*
   **Author**: We have now added the Equations to the Figures

**Reviewer**: *Technical suggestion: Because of the fraction you could put the definitions not in the figure caption but in the paper text where there is more space to introduce them.*
**Author response**: We have now removed the Equations from the Figure captions, and instead added it to the text, together with an additional explanation of the parameters.

4. **Reviewer (first review):** *Section 4.2: It would be good if the subclusters can at least be described a bit more in their configuration. Furthermore, the discussion should also include the results from a baseline model that is not optimized for comparison.* **Author**: A Figure has been added that shows the coordinates of the wind turbines within the farm.
**Reviewer**: *An answer to the second part of the comment was given to the other reviewer. The authors have decided against comparing their results to an uncalibrated/different model. In my opinion this is of critical importance in the future to benchmark the proposed method.*
**Author response**: We have decided to incorporate a comparison of results against the uncalibrated model. This comparison can be observed in Figure 15, with some additional text explaining the error metric (see author response 3 to the associate editor).

**Reviewer 2:**

1. **Reviewer:** *I was not satisfied with the authors' response to the first general comment of Reviewer 2. The found relationship between the wake growth rate and the turbulence intensity differs from a body of literature that observed a stronger relationship from field experiments (e.g. Trabucchi et al., 2017), wind tunnel observations (Ishihara and Qian, 2018\*), and large-eddy simulations (e.g. Niayifar and Porte-Agel, 2016). In my opinion, a discussion of what might contribute to the differences and mentioning possible limitations of the present study is required in the manuscript given the differences.*

   **Reviewer:** *The authors replied to the general comment no. 1 of Reviewer 2 that their 'decision to focus on the velocity deficit model and to exclude others, like the wake-added turbulence model, comes from the interdependencies these models show. Including both submodels could lead to a multimodal solution space, where certain parameters in one submodel influencing adjustments in another." From this, I understand that the optimized parameters might change if those submodels are changed. Therefore, the found wake growth rates and the conclusion that 'comparing the optimized parameters to the baseline reveals that the baseline parameters underestimate the wake effects, which subsequently leads to an overestimation of the expected yield' should be softened by mentioning the limitation in the paper explicitly in my opinion.*

   **Reviewer:** *However, there are also possible reasons that can be mentioned why it might be different from the studies cited in the first paragraph above. They only used isolated wind turbines to determine the relationship between turbulence intensity and wake growth rate. It might be that the relationship is naturally different inside a wind farm, because the turbulence of the interior wind farm flow changes not only in its intensity, but also in scales. Some papers showed that the thrust coefficient of the wind turbine affects the wake recovery as well.*

**Author response:** We acknowledge the limitation arising from our decision to analyze the wind farm at a single TI of 0.06. The choice to go for a TI of 0.06 was based on the work by Doekemeijer B. M. (2022), but we recognize that this assumption will make the results only representative for a given TI of 0.06, meaning that changing the TI in the model will require additional calibration. We have incorporated this to both the optimization results section and the conclusion section.

Moreover, we are aware of the differences between the above-mentioned papers and our acquired results. Our analysis focuses on the collective behavior of the wind farm, where the mentioned studies focus on a single turbine wake, where the focus is on a given number of rotor diameters behind the turbine. Tuning on such scale can make the tuning parameters account for flow physics, which are inherently different to those compared to the single wake case.

Nevertheless, assuming a TI equal to 0.06 is a conservative estimate for AEP calculations compared to other available options, such as calculating TI from SCADA, met mast measurements, or by using IEC standards as reference, which could lead to larger deviations from the reference value determined by Niayifar and Porte Agel (2016). Our focus is on quantitatively optimizing the performance of the entire wind farm, rather than qualitatively analyzing the wake effect of an isolated wind turbine. The effect of turbulence is something we are willing to study in the future.

In conclusion, we recognize the limitations associated with our chosen TI and the observed differences between our findings and the reference literature and we have now addressed this within our paper, both in the optimization results section and the conclusion section.

**Reviewer:** *Regarding the implementation of the model that was optimized here, the reply also did not address my initial concern. It is true that nacelle mounted anemometers are not perfect, which can be seen from a comparison to upstream looking lidars/upstream located met towers that have been published for some field campaigns. However, the power curve provided by the manufacturer is of unknown quality (or at least it is not provided in the manuscript) and the power curve might have been created for an undisturbed inflow (e.g. an inflow that can be described with a log- or power-law). It remains the question if and how it can be applied to a waked or partially waked wind turbine in the interior of a wind farm. The reply of the authors did not rule out that the differences in wake growth rates found here and those in other literature might be partly explained by such issues. Therefore, I am missing here as well that the author mention such possible limitations of their findings.*

**Author response:** We have refined the power curve filtering process, as visualized in Figure 3, by narrowing the filtering boundaries. While I do agree that the power-curve might have been developed assuming an undisturbed inflow, it is still unclear which approach more accurately estimates wake losses. For instance, in scenarios where a wind turbine experiences a small partial wake effect on either its left or right side, the impact on the nacelle-mounted anemometer is expected to be minimal, yet a reduction in power production is expected. In our opinion, a comprehensive analysis of such cases

would necessitate a parallel study with the cost-function based on wind speed rather than power production. Nevertheless, we will indeed acknowledge this limitation, along with the previously mentioned ones, in our paper.

2. **Reviewer:** *Section 2.4 / line 252 – 259: The filtering criterion is still mysterious to me and the clarification by the authors have not improved this. The authors say that they sometimes observe discrepancies between the active power and the power set point at wind speeds above rated wind speeds. Addressing this issue with threshold for the ratio of the wind speed variance is still not intuitively clear to me. I suspect that the criterion might work for the authors special circumstances at their experimental site, but I doubt whether it is a universal approach to detect this issue with the turbine operation (I wanted to test this on a set of SCADA data to verify but I did not find the time in the end). Therefore, I believe the authors should add the following information to the paper:*
   (1) *A summary of the impact of this filter criterion (How many data points does it remove from the total? Can it be shown that that it is successful at removing what the authors describe?).*
   (2) *It should be clearly stated whether the filter criterion is based purely on a correlation or whether there is also a physical causation to support it (and in case of the former that it might not be a procedure that can be universally recommended to identify this state of turbine operation).*

**Author response:**

We identified that this additional filtering criteria was necessary due to either above-rated curtailment, reaching values close to rated power production, or minor underpredictions of power prediction at rated capacity. It became clear that a one-size-fits-all approach might not be suitable for every wind farm, therefore we tried to explore alternative methods. Consequently, we have refined the power-curve filtering process to more precise exclude data points related to curtailment near the rated power. This can be observed in Figure 3.

References:

Doekemeijer, B. M., Simley, E., Fleming, P.: Comparison of the Gaussian Wind Farm Model with Historical Data of Three Offshore Wind Farms, Energies, 15, 1964, https://doi.org/10.3390/en15061964, 2022

Nygaard, N. G., Steen, S. T., Poulsen, L., and Pedersen, J. G.: Modelling cluster wakes and wind farm blockage, Journal of Physics: Conference Series, 1618, 062-072, https://doi.org/10.1088/1742-6596/2265/2/022008, 2020

---

## Author Response (AR3)

**Reply to Editor:**

Thank you for your report. We have revised the abstract to explicitly mention the limitations of the work together with a clear statement advising against the direct application of the calibrated tuning parameters to different sites without further recalibration.